# Postcranial osteology of *Beipiaosaurus inexpectus* (Theropoda: Therizinosauria)

**Chun-Chi Liao**[1,2,3], **Lindsay E. Zanno**[4,5], **Shiying Wang**[1,2,3], **Xing Xu**[1,2]*

**1** Key Laboratory of Vertebrate Evolution and Human Origins of Chinese Academy of Sciences, Institute of Vertebrate Paleontology and Paleoanthropology, Chinese Academy of Sciences, Beijing, China, **2** CAS Center for Excellence in Life and Paleoenvironment, Beijing, China, **3** University of Chinese Academy of Sciences, Beijing, China, **4** Paleontology, North Carolina Museum of Natural Sciences, Raleigh, North Carolina, United States of America, **5** Department of Biological Sciences, North Carolina State University, Raleigh, North Carolina, United States of America

* xu.xing@ivpp.ac.cn

**Data Availability Statement:** All relevant data are within the manuscript.

**Funding:** This study was supported by the National Natural Science Foundation of China (41688103) and the scholarship provided by the University of

## Abstract

*Beipiaosaurus inexpectus*, from the Lower Cretaceous Yixian Formation (Sihetun locality, near Beipiao), Liaoning, China, is a key taxon for understanding the early evolution of therizinosaurians. Since initial publication in 1999, only the cranial elements of this taxon have been described in detail. Here we present a detailed description of the postcranial skeletal anatomy of the holotype specimen of *B. inexpectus*, including two never before described dorsal vertebrae from the anterior half of the series. Based on these observations, and comparisons with the postcranial skeleton of therizinosaurian taxa named since the most recent diagnosis, we revised the diagnostic features for *B. inexpectus* adding three new possible autapomorphies (PII-3 shorter than PIII-4, subequal length of the pre- and postacetabular portions of the ilium, and equidimensional pubic peduncle of ilium). Additionally, we also propose three possible synapomorphies for more inclusive taxa (Therizinosauroidea and Therizinosauridae) and discuss implications for evolutionary trends within Therizinosauria. The newly acquired data from the postcranial osteology of the holotype specimen of *B. inexpectus* sheds light on our understanding of postcranial skeletal evolution and identification of therizinosaurians.

## Introduction

Therizinosaurus were a rare and bizarre clade of maniraptoran dinosaurs. Late-branching taxa possessed characteristic features such as a rostrally endentulous snout, dentary with a lateral shelf, small tightly packed lanceolate teeth, diminutive skulls, elongate neck, extremely broad and opisthopubic pelvis, shortened tibia, and tetradactyl pes [1–3]. Skeletal remains have been discovered from Cretaceous strata of Asia and North America [4–8], although *Eshanosaurus deguchiianus* from the Lufeng Formation, Yunnan, China might represent an Early Jurassic therizinosaurian [9, 10].

*Beipiaosaurus inexpectus* is an early-branching therizinosaurian from the Lower Cretaceous Yixian Formation in Liaoning, China. As one of the earliest-branching members of the clade,

Chinese Academy of Sciences (UCAS) to C-C L. The funders had no role in study design, data collection and analysis, decision to publish, or preparation of the manuscript.

**Competing interests:** The authors have declared that no competing interests exist.

the discovery of this taxon was key for disentangling the broader phylogenetic relationships of therizinosaurians. Particularly, because it lacks many features in late-branching members of the clade that are convergent with sauropodomorphan dinosaurs such as a tetradactyl pes and broad proximal first metatarsal as well as providing direct evidence of the presence of filamentous feathers in this clade [8]. Despite the significance of *B. inexpectus* for disentangling early anatomical evolution in the clade, detailed morphological descriptions are restricted to the cranial elements, select elements of the appendicular skeleton, the "pygostyle," and a unique type of feathers called elongated broad filamentous feathers (EBFFs) [8, 11–13]). A detailed description of the postcranial skeleton of the specimen is needed to further understand this important taxon. Here we present such a description of the postcranial skeletal osteology of the holotype specimen (IVPP V11559) in order to fill in new anatomical information on this key taxon and shed light on the evolution of the postcranial skeleton within Therizinosauria.

## Methods

We follow the phylogenetic terminology and definitions of Zanno [3] for Therizinosauria, Therizinosauroidea, and Therizinosauridae. In contrast to most paleontological literature, we identify the three tetanuran manual digits as II-III-IV following the ornithological literature and some recent paleontological studies in light of avian digit homology research [14, 15]. Photography of *B. inexpectus* was performed in the Institute of Vertebrate Paleontology and Paleoanthropology. In each figure, the matrix and neighbouring bones were recolored to highlight the bone in Adobe Photoshop CC 2018. No permits were required for the described study, which complied with all relevant regulations.

### Institutional abbreviations

IVPP, Institute of Vertebrate Paleontology and Paleoanthropology, Beijing, China.

### Systematic paleontology

Dinosauria Owen, 1842

Theropoda Marsh, 1881

Coelurosauria sensu Gauthier, 1986

Therizinosauria Russell, 1997

Therizinosauroidea Russell and Dong, 1993

*Beipiaosaurus inexpectus* Xu, Wang et Tang, 1999

**Holotype** IVPP V11559, a partial, semi-articulated skeleton including some cranial elements and most of the postcranial elements. (Note: After *Beipiaosaurus inexpectus* was named in 1999, additional materials of the holotype [IVPP V11559] were collected from the type locality [11]. The holotype specimen thus comprises more skeletal elements than listed in the first description.)

 **Type locality and horizon** Sihetun locality near Beipiao, Liaoning, China. Lower Cretaceous (Aptian) Yixian Formation [16].

 **Revised diagnosis.** A small therizinosaurian possessing the following autapomorphies among Therizinosauria (newly added features noted with an asterisk*): postorbital process of frontal large and abruptly transits from orbital rim [13]; parietal with a long and sharp anterior process [13]; ventral ramus of parietal squamosal process extremely long [13]; and external

mandibular fenestra deep dorsoventrally and extremely posteriorly located [13]; four fused caudal dorsals [3]; pygostyle incorporating up to seven caudal vertebrae [3]; "rectangular buttress" on MCII expressed as a triangular flange [3]; elongate lateral articular surface on manual phalanx II-1 [8]; manual ungual of digit II shorter than digit III*; subequal length of pre- and post-acetabular process of ilium*; equidimensional pubic peduncle of ilium (anteroposterior width subequal to mediolateral width)*; obturator process of ischium sinusoidal, with ventrally deflected distal portion [3]; ischial boot approximately twice anteroposterior depth of distal shaft [3]; low ridge on anterior femoral shaft extending proximally from medial condyle [3].

## Description and comparisons

IVPP V 11559 is considered a skeletally immature individual because of the lack of fusion in the non-cranial axial skeleton including cervical vertebrae and corresponding cervical ribs, dorsal, sacral, and most caudal vertebral centra and their corresponding neural spines (except for posterior caudals) [17, 18]. Postcranial skeletal elements that variably fuse during ontogeny in theropods such as the ipsilateral scapula and coracoid, distal carpals II and III (known to at least partially fuse later in ontogeny in *Falcarius*; [19], and the distal tarsals and tibia are also unfused.

### Axial skeleton

IVPP V11559 preserves four cervical, six dorsal, three sacral, and 30 caudal vertebrae (Figs 1–3 and Table 1). Most therizinosaurians except *Jianchangosaurus* and *Nanshiungosaurus* are represented by incomplete presacral vertebral columns [3, 20, 21]. There are 10 cervicals, 12 dorsals, and five sacrals in *Jianchangosaurus* [20] and 11 cervicals, 10 dorsals, and five sacrals in *Nanshiungosaurus* [22], although a more recent study suggests that *Nanshiungosaurus* ('brevispinus', NIGP V4731) has six sacrals [3].

**Cervicals.** Four incomplete cervical vertebrae are present, exposed in dorsal and lateral view (Fig 1A and 1B), and all of them are poorly preserved. Given that these cervical vertebrae are not elongate, they likely derive from the posterior cervical series. The total number of cervicals for *B. inexpectus* is unknown, but based on the closely related taxa *Beipiaosaurus* sp. (STM 31–1) [12] and *Jianchangosaurus* [20], we estimate around nine or 10. The later-branching therizinosaurians, 'Nanshiungosaurus' bohlini [22] and *Neimongosaurus* [23]), have 11–12 and 14 cervicals, respectively. An elongate neck with more than 10 cervicals is also seen in closely related oviraptorosaurians such as *Caudipteryx*, *Citipati*, and *Khaan* [24, 25].

In dorsal view, the neural arch is wider than its corresponding centrum, and both the pre- and postzygapophyses are lateral to the centrum, as in other therizinosaurs [19, 23, 26]. The neural spines of the cervicals are extremely low, anteroposteriorly short [8], and mediolaterally compressed with a straight dorsal border, in lateral view. Low and undeveloped cervical neural spines are a characteristic trait observed in many other therizinosaurians, such as *Nothronychus* [27], *Neimongosaurus* [23], *Jianchangosaurus* [20], and *Falcarius* [19]. The neural spine is centered on the neural arch, making an 'X'-shape in dorsal view.

The articular surfaces of the postzygapophyseal facets face ventrally and are rounded. The prezygapophyses are robust and bear oval-shaped articular surfaces, similar to *Jianchangosaurus* [20]. In dorsal view, there is an intraprezygapophyseal lamina between the prezygapophyses, which only reaches the posterior margin of the articular facets. Between the postzygapophysis, there is a web of bone connecting the two parts called the intrapostzygapophyseal lamina (sensu Wilson, 1999 [28]), which is seen in other therizinosaurians such as *Falcarius* and *Neimongosaurus*, and is present on a broad distribution of coelurosaurs including ornithomimids, dromaeosaurs, troodontids, and oviraptorosaurs [19]. This lamina restricted

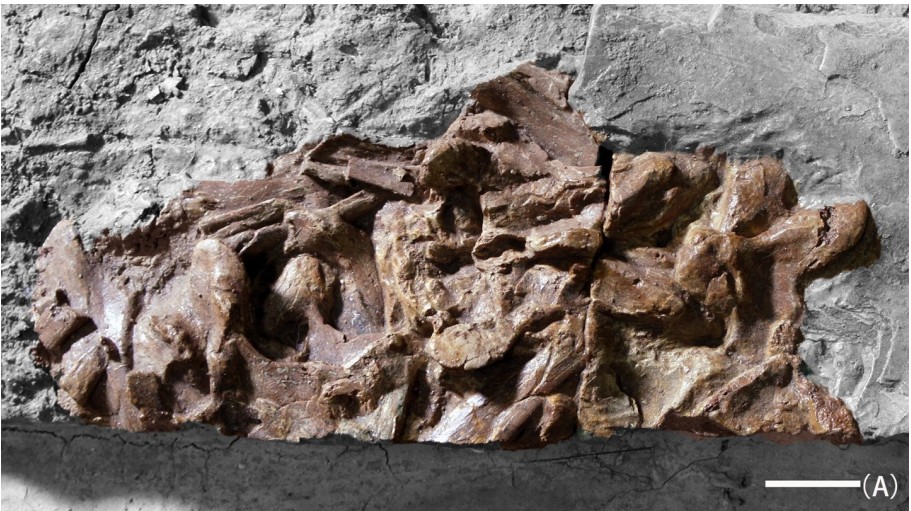

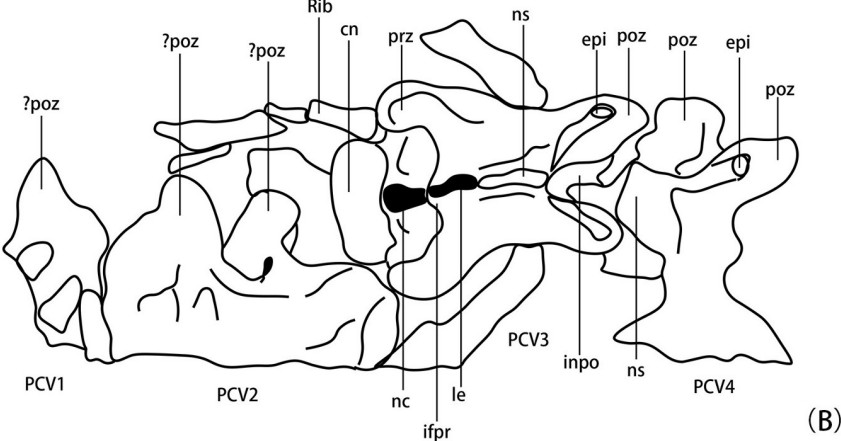

**Fig 1.** Cervical vertebrae of *B. inexpectus* (IVPP V 11559) in dorsal view; A. digitally enhanced photograph; B. interpretive line drawing. **Abbreviations: cn**, centrum; **epi**, epipophyses; **le**, ligamentum elastici; **ifpr**, infraprezygapophyseal lamina; **inpo**, interpostzygapophyseal lamina; **ns**, neural spine; **PCV1-3**, posterior cervical vertebrae 1–3; **poz**, postzygapophyses; **prz**, prezygapophyses. Scale bar equal 2 cm.

the ligamentum elastici to a small fossa at the base of the neural spine (the interspinous fossa). As in *Falcarius* and *Jianchangosaurus*, there are prominent epipophyses above the postzygapophysis [19, 20].

The cervical centra of *B. inexpectus* are amphicoelous. Most therizinosaurians have amphicoelous cervical centra (e.g., *Falcarius* [19]; *Martharaptor* [29]; *Jianchangosaurus* [20]; *Neimongosaurus* [23]; and *Nothronychus* [27]); or amphiplatyan centra (*Alxasaurus* [6]; and *'Nanshiungosaurus' bohlini*, [30]). The preserved third posterior cervical centrum is only slightly longer than the longest preserved posterior dorsal centrum (3%). However, despite shortening of the cervical centra posteriorly, all cervical centra remain anteroposteriorly longer than the preserved dorsal centra in *B. inexpectus*, as in other therizinosaurs. In *Neimongosaurus*, the penultimate cervical centrum is 17% longer than the longest dorsal centrum [23], and it is 25% and 12.5% longer in larger and smaller individuals, respectively in *Alxasaurus* [6].

Ribs are not fused to posterior cervicals, an ontogenetically variable feature that might strictly reflect the early ontogenetic stage of *B. inexpectus*. This condition is also seen in

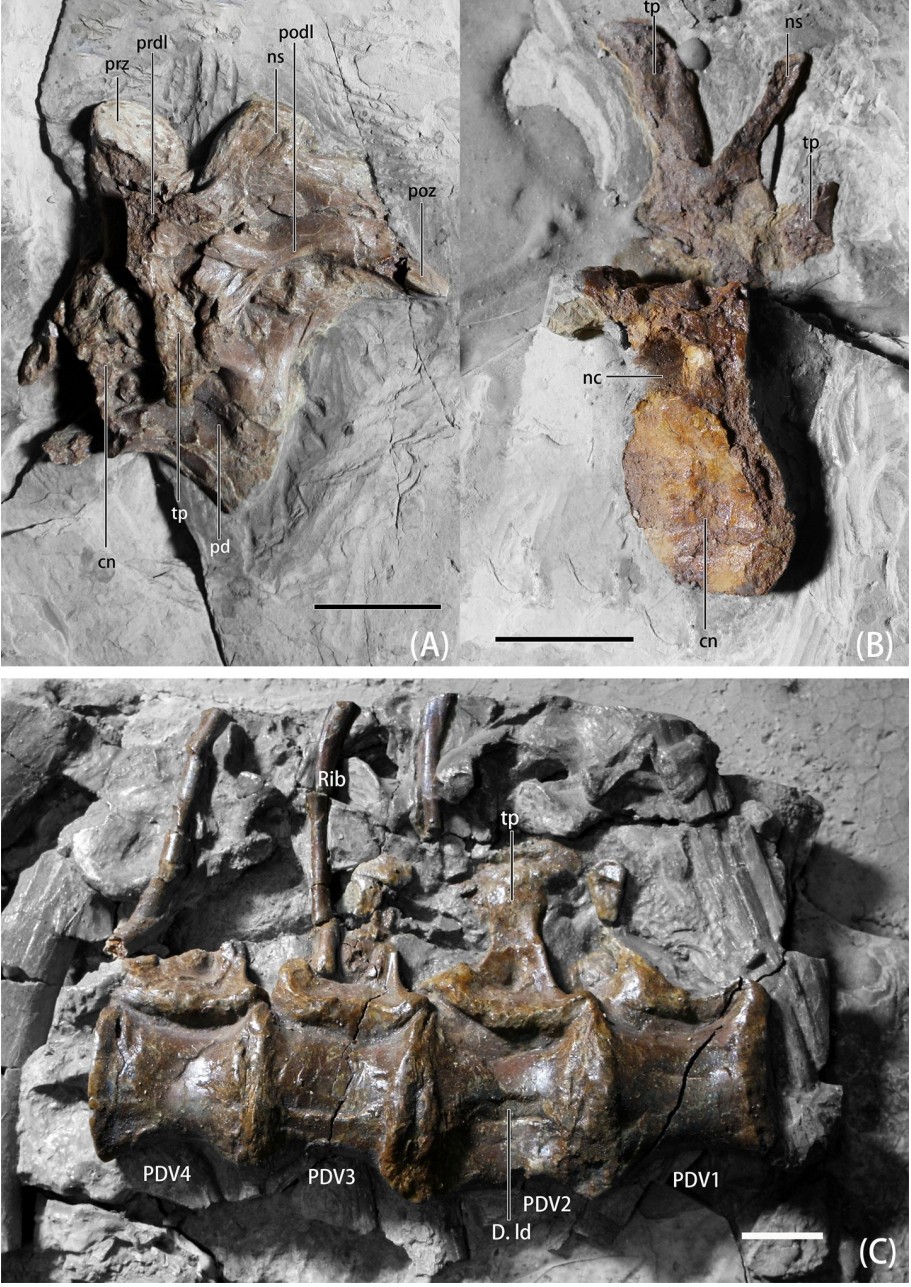

**Fig 2.** Photographs of the dorsal vertebrae of *B. inexpectus* (IVPP V 11559) A. antero-middle dorsal in lateral view; B. middle dorsal in anterolateral view; C. posterior dorsals in lateral view. **Abbreviations: cn**, centrum; **D. ld**, lateral depression of dorsals; **nc**, neural canal; **ns**, neural spine; **pd,** pneumatic depression; **PDV1-4,** posterior dorsal vertebrae 1–4; **podl**, postzygadiapophyseal lamina; **poz**, postzygapophyses; **prdl**, prezygadiapophyseal lamina; **prz**, prezygapophyses; **tp**, transverse process. Scale bars equal 2 cm.

*Jianchangosaurus* [20] and *B.* sp. [12]. Cervical ribs are slender and long, longer than corresponding centra. Ribs elongation is shared with *B.* sp. [12] and *Jianchangosaurus* [20], but contrasts with *Neimongosaurus* [23] and *Alxasaurus* [6], which bear short and broad cervical ribs.

**Dorsals.**   Six dorsals are preserved, four of them are preserved in articulation and exposed in lateral view (Fig 2C). These vertebrae were figured in Xu et al. ([8], Fig 1). The remaining

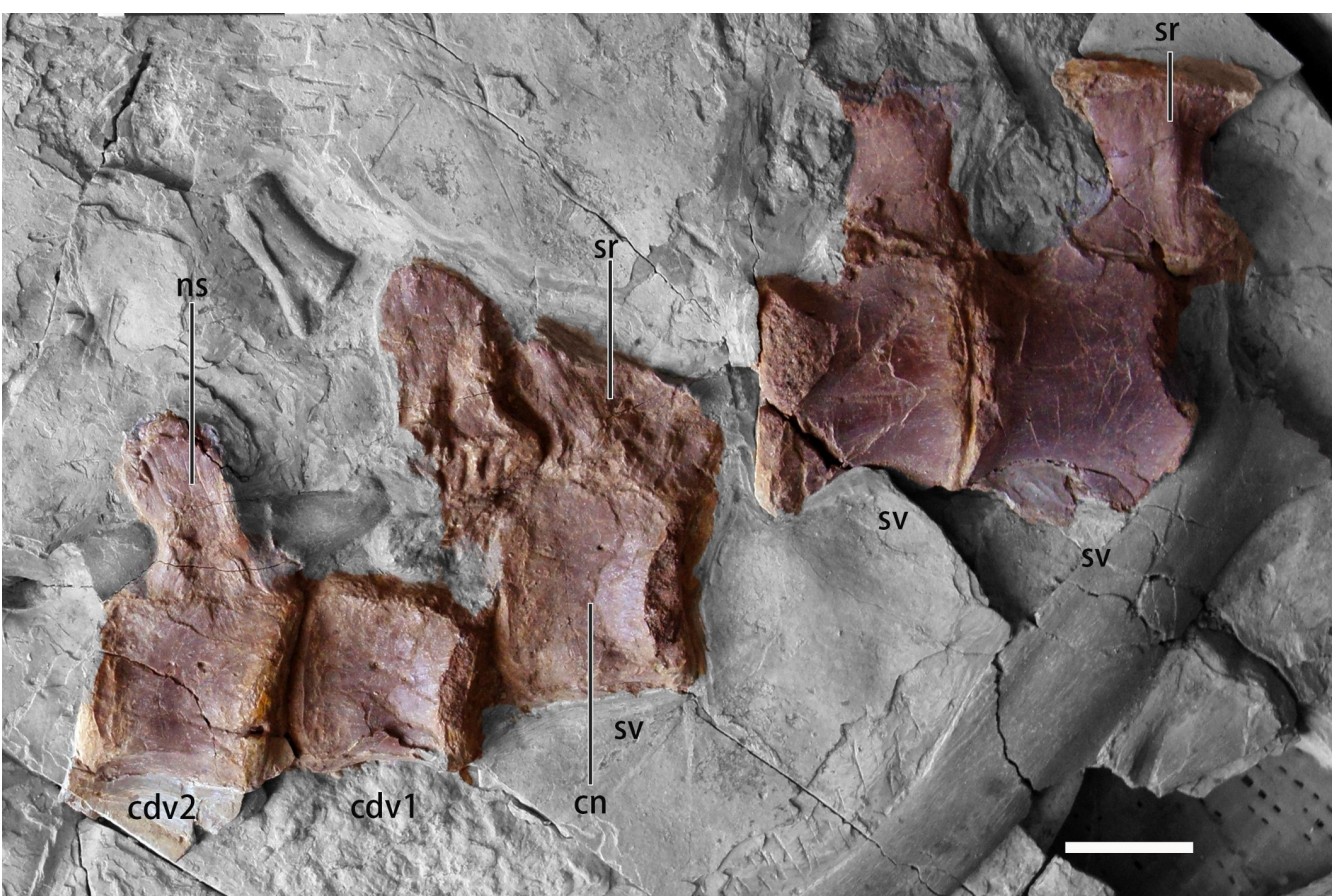

**Fig 3. Photographs of sacral and anterior caudal vertebrae of *B. inexpectus* (IVPP V 11559). Abbreviations: cn,** centrum; **cdv1-2,** caudal vertebrate 1–2; **ns,** neural spine; **sv**, sacral vertebrate; **sr**, sacral rib. Scale bars equal 2 cm.

two preserved dorsal vertebrae are preserved in anterolateral and lateral view (Fig 2A and 2B) and were discovered isolated along with additional materials of the holotype specimen and published in Xu et al. [11]. The four articulated dorsals are interpreted as deriving from the posterior portion of the series, and the isolated two dorsals are interpreted as deriving from the anterior to middle portion.

Neural sutures appear to be at least partially fused in the more middle dorsals (Fig 2A and 2B). The one preserved in lateral view is interpreted as an antero-middle dorsal based on its lamination, the low neural spine, arched postzygapophyses, and the longer centrum (Fig 2A). The lamina and fossa system are developed, yielding pre- and postzygadiapophyseal laminae. The well-developed lamina and fossa system is also a feature that can be seen in many therizinosaurians, including *Falcarius* [19], *Jianchangosaurus* [20], *Lingyuanosaurus* [21], *Shuzhousaurus* [30], *Alxasaurus* [6], *Neimongosaurus* [23], and *Nanshiungosaurus* [22]. In lateral view, the centrum is sub-rectangular in shape, and the ventral border is concave slightly, making the articular facets relatively expanded dorsoventrally. There is a depression on the lateral surface, positioned under the transverse process. The same feature can also be seen in the posterior dorsals.

The vertebra preserved in anterolateral view is interpreted as a middle dorsal based on the height of the neural spine and upswept angle of the transverse processes (Fig 2B). The shape of the neural spine is gracile, and its dorsal surface is slightly expanded anteroposteriorly. In

**Table 1. Measurements in millimeters of vertebrae of *B. inexpectus* (IVPP V 11559).**

| Cervical vertebrae (Fig 1) | Width, maximum | Length, maximum |
|---|---|---|
| Posterior cervical 1 | - | - |
| Posterior cervical 2 | - | - |
| Posterior cervical 3 | - | 46.3 |
| Posterior cervical 4 | - | - |
| **Dorsal vertebrae (Fig 2)** | **Length, maximum** | **Hight, maximum** |
| Antero-middle dorsal (Fig 2A) | 44.0 | 47.2 |
| Middle dorsal (Fig 2B) | - | - |
| Posterior dorsal 1 | 44.9 | - |
| Posterior dorsal 2 | 40.9 | - |
| Posterior dorsal 3 | 37.8 | - |
| Posterior dorsal 4 | 37.1 | - |
| **Sacral vertebrae (Fig 3)** | | |
| Antepenultimate sacral | 38.6 | - |
| Penultimate sacral | 36.7 | - |
| Posteriormost sacral | 32.2 | - |
| **Anterior to middle caudal vertebrae (Figs 3 and 4A)** | | |
| 1 | 31.8 | - |
| 2 | 31.9 | 62.4 |
| 3 | - | - |
| 4 | - | - |
| 5 | - | - |
| 6 | - | - |
| 7 | 34.2 | 39.8 |
| 8 | 33.1 | 40.1 |
| 9 | 33.1 | 39.7 |
| 10 | 33.6 | 37.6 |
| 11 | 33.0 | - |
| **Posterior caudal vertebrae (Fig 4B)** | | |
| Posterior caudal 1 | - | - |
| Posterior caudal 2 | 27.1 | 23.5 |
| Posterior caudal 3 | 26.5 | 25.3 |
| **Posteriormost caudal vertebrae (Fig 4D)** | | |
| Posteriormost caudal 1 | 17.7 | 12.6 |
| Posteriormost caudal 2 | 16.6 | 13.7 |
| Posteriormost caudal 3 | 15.4 | 12.1 |
| Posteriormost caudal 4 | 14.3 | 10.9 |
| Posteriormost caudal 5 | 12.2 | 10.9 |
| Posteriormost caudal 6 | 10.8 | 8.6 |
| Posteriormost caudal 7 | 8.9 | 9.5 |
| Posteriormost caudal 8 | 5.5 | 7.1 |
| Posteriormost caudal 9 | 6.3 | 5.6 |
| Posteriormost caudal 10 | - | 4.4 |
| Posteriormost caudal 11 | - | - |
| Posteriormost caudal 12 | 3.0 | - |
| Posteriormost caudal 13 | 2.1 | 3.6 |

comparison with the antero-middle dorsal, the neural spine of this vertebra is dorsoventrally taller and approximately the same dorsoventral height as the centrum. In this respect, the

neural spine of *B. inexpectus* is more similar to that of later-branching therizinosaurians such as *Nothronychus* [31], *Alxasaurus* [6], '*Nanshiungosaurus*' *bohlini* [22], *Neimongosaurus* [23], and *Erliansaurus* [32], which possess relatively taller neural spines. Whereas in other earlier-branching members such as *Falcarius* [19], *Jianchangosaurus* [20], and *Lingyuanosaurus* [21], dorsal neural spines are robust, anteroposteriorly broaden, and reduced in height. The neural canal in anterior view is pentagon-shaped and dorsoventrally tall. Parapophyses are not prominent, and are present as a depression on the anterior rim, flush with neural arch. In anterior view, the articular facets of the centra are oval in shape and dorsoventrally high. Articular facets are slightly concave, suggesting an amphicoelus condition.

Neural sutures are unfused in the preserved posterior dorsals (Fig 2C), and may also be incompletely fused in the more anterior dorsals (Fig 2A and 2B). The neural arches of the four articulated dorsals are crushed and the neural arch is exposed only in ventral view. The transverse processes are elongate (i.e., the mediolateral length of the transverse process is approximately equal to the dorsoventral height of the corresponding centrum). In ventral view, the transverse process narrows in mid-shaft and slightly expands anteroposteriorly at the distal-most point. Parapophyses are not prominent, and present as a depression on the anterior rim, flush with neural arch. Posterior dorsal centra are spool-shaped generally and subrectangular in lateral view, as in *Falcarius* [19] and *Neimongosaurus* [23]. Centra are apneumatic, but do bear a lateral depression.

Ribs of dorsals are incomplete. However, it can be observed that they are elongate, extremely slender, and subtriangular in cross-section (Fig 4C).

**Sacrals.** The total number of sacrals in *B. inexpectus* is currently unknown. The last three sacrals are preserved in the material found within the re-excavation. The antepenultimate and the penultimate sacrals are fused. However, the posteriormost centrum is separated. All of the sacral centra are preserved in ventral view with only the dextral sacral ribs attached (Fig 3). The centra of the posteriormost three sacrals are subrectanglar in lateral view, unlike the sacrals of *Falcarius*, which are strongly medially constricted producing an hourglass-like shape in ventral view [19]. The sacral ribs of the three posteriormost sacrals are oriented laterally and expand anteroposteriorly. They are robust and their length is subequal to the width of their centra.

**Caudals.** The majority of caudal vertebrae were recovered among the second set of materials published in 2003 (Figs 3 and 4A and 4B, 4D and 4E). These constitute a nearly complete tail with 30 caudals [11]. However, some caudal vertebrae are not preserved, thus the total number of caudals of *B. inexpectus* was more than 30. The early-branching therizinosaurian, *Falcarius* is estimated to have about 35 caudal vertebrae [19]. In later-branching therizinosaurians, such as *Neimongosaurus* [23] and *Alxasaurus* [6] an incomplete series of 22 and 21 caudals are preserved, respectively.

Most caudal vertebrae are transversely compressed and exposed in lateral view. All of the caudal centra are apneumatic, amphicoelous, and constricted medially and ventrally. The anterior caudal centra are relatively robust with a subequal height to length ratio. Caudal centra gradually become anteroposteriorly elongate, as in other early-branching therizinosaurians (e.g., *Falcarius* [19]), as well as theropods generally. This condition is different from that observed in late-branching therizinosaurians, such as *Alxasaurus* [6] and *Northychus* [31], which possess anteroposteriorly abbreviated distal caudals. Unlike in other portions of the vertebral column, the neural arches and centra are fused in the caudals.

Neural spines in the anterior portions of the caudal series are subequal to the height of centra and reduce in height posteriorly. On the preserved first through third neural spines, the neural spines are less anteroposteriorly expanded, slightly posterodorsally oriented, and possess a convex dorsal margin. The neural spines become more box-shaped in lateral view and also shorten in anteroposterior length, maintaining their relative proportions. Neural spines of

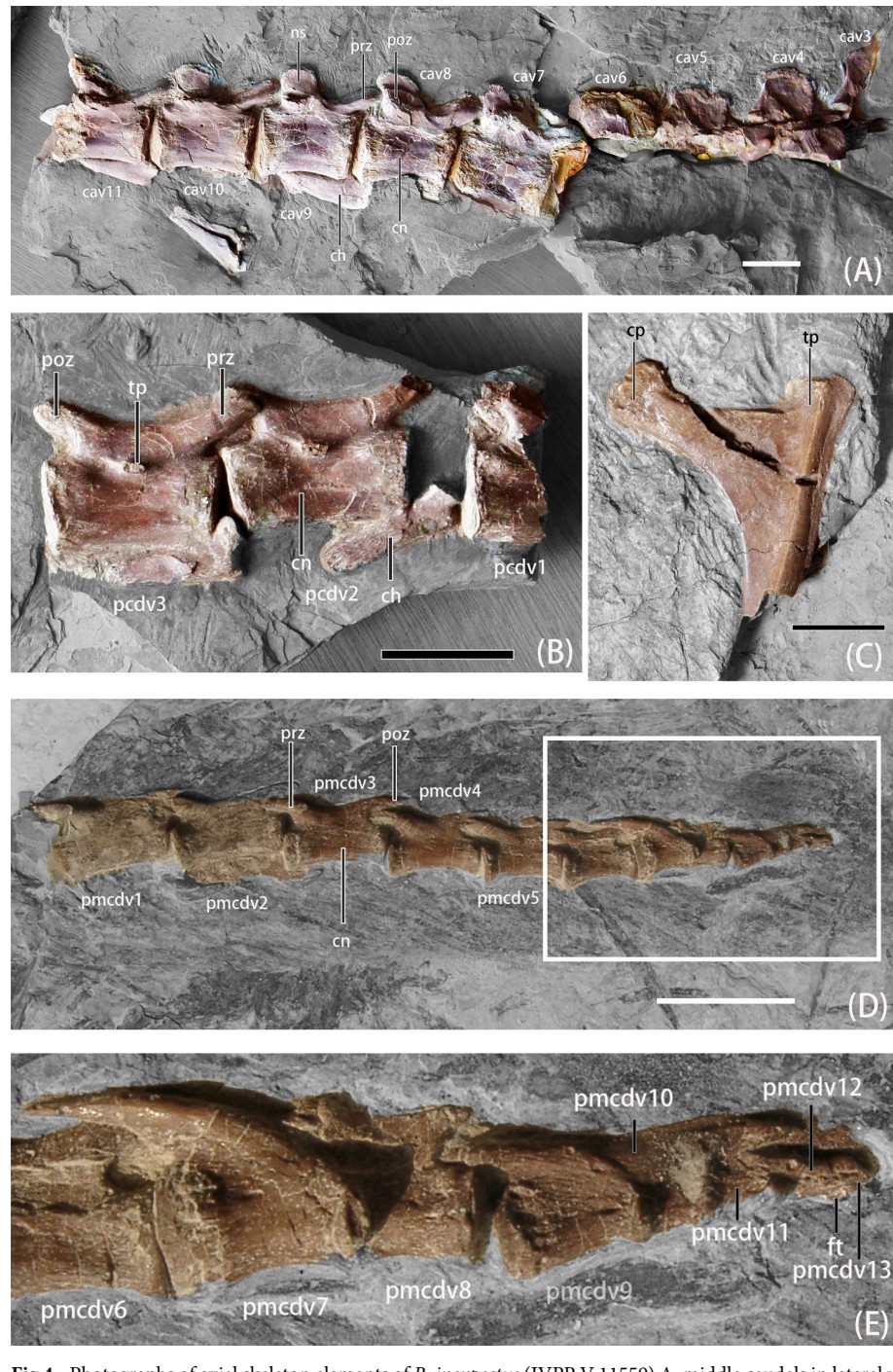

**Fig 4.** Photographs of axial skeleton elements of *B. inexpectus* (IVPP V 11559) A. middle caudals in lateral view; B. posterior caudals in lateral view; C. proximal end of dorsal rib in lateral view; D. posterior caudals with pygostyle-like structure; E. close-up for pygostyle-like structure. **Abbreviations: cn,** centrum; **cav3-11,** caudal vertebrae 3–11; **ch,** chevron; **cp**, capitulum; **ft**, fusion tubercle; **ns**, neural spine; **pcdv1-3**, posterior caudal vertebrae 1–3; **pmcdv 1–13**, posteriormost caudal vertebrae 1–13; **poz**, postzygapophyses; **prz**, prezygapophyses; **tb**, tuberculum; **tp**, transverse process. Scale bars equal 2 cm.

mid-caudal vertebrae are more posteriorly positioned, becoming flush with the postzygapophysis (Fig 4A). Further posteriorly, neural spines are absent (Fig 4B).

The pre- and postzygapophyses are steeply elevated and posteriorly inclined, as in most theropods generally, but more prominently in therizinosaurians and oviraptorosaurians [3]. In anterior caudal vertebrae, the prezygapophyses are more robust with oval articulation facets, and gradually become more slender and tapering distally. Prezygapophyses are elongate, approximately half the length of the neural arch and longer than the postzygapophyses. Postzygapophyses are mostly crushed or not visible in anterior to mid-caudal vertebrae. In posterior vertebrae, the postzygapophyses remain relatively short. The transverse processes are mostly crushed or incomplete. When present, they reduce from anteroposteriorly elongate and dorsoventrally thin processes to only small tubercles at the base of the middle neural arches along the axis. Posteriorly, the transverse processes become more anteriorly positioned.

The five posteriormost caudal vertebrae are fused together in a pygostyle-like structure (Fig 4D and 4E) [11], with ventrally concave and dorsally convex margins, as well as a blunt posterior end. A prominent line of fusion is still visible between the anterior two centra on the "pygostyle," whereas the last two caudal vertebrae are completely fused, and can only be distinguished by a well-developed tubercle (Fig 4E). The neural spines of these five caudals are fused, and the margins are obscured. The phylogenetic distribution of this feature in other therizinosaurians is currently unknown, because the posteriormost portion of the caudal series is not definitively preserved in other therizinosaurians. When compared to the pygostyle-like structures exhibited by oviraptorosaurs, the "pygostyle" of *B. inexpectus* appears morphologically more similar to that of birds, in being more co-ossified and possessing stronger dorsally curved axes [11, 33].

**Chevrons.**   Some chevrons are preserved in articulation with the caudals; they are approximately the same length as the centra with which they once articulated (Fig 4A and 4B). The proximal articular facets are concave in dorsal view, with a more extended posterior process. The shafts of the chevrons are straight as in *Neimongosaurus* [23], but different from the morphology of *Falcarius* and *Alxasaurus*, which are posteriorly deflected. Some of them possess an anterior tuberosity [19].

## Pectoral girdle

IVPP V11559 preserves the right scapula, both coracoids, and a partial furcula with its corresponding impression (Fig 5 and Table 2).

**Furcula.**   A partial right epicleidium of the furcula and its exquisite impression are preserved. Together, these demonstrate that it is a widely arched bone with an oblate-shaped cross-section (Fig 5B) [8]. Although Xu et al. [8] noted the absence of the hypocleidium on the furcula of *B. inexpectus*, Zanno [19] suggested the incomplete impression of the furcula and poor development of this feature in other taxa (i.e., *Falcarius*, [2]; *Nothronychus* [31]), makes this absence uncertain. In therizinosaurians, only *Neimingosaurus* lacks the development of hypocleidium entirely to date [19, 23].

The intraclavicular angle of *B. inexpectus* is ~145 degrees, similar to that of *Neimongosaurus*, which is about 135 degrees [23, 31], and unlike that of *Falcarius*, which is more acute (~104 degrees [2]). The epicleidea of *B. inexpectus* is straight, not bowed at its termini. In general shape, the furcula of *B. inexpectus* most closely resembles that of *Neimongosaurus* [23], as opposed to *Nothronychus* and *Falcarius*, which display at least some lateral deflection of the epicleidal termini [19, 31]. The distal termini of *B. inexpectus* tapers to a point, as in *Jianchangosaurus*, and unlike in *Neimongosaurus*, which bears a robust epicleidium with blunt distal ends [23]. In general, the furculae of therizinosaurians including *B. inexpectus*, *Falcarius*, *Nothronychus*, and *Neimongosaurus* are gracile and V-shaped.

**Coracoid.**   Both coracoids are preserved, yet incomplete. The right element is more complete than the left element (Fig 5A). The coracoids of *B. inexpectus* are subrectangular, which is

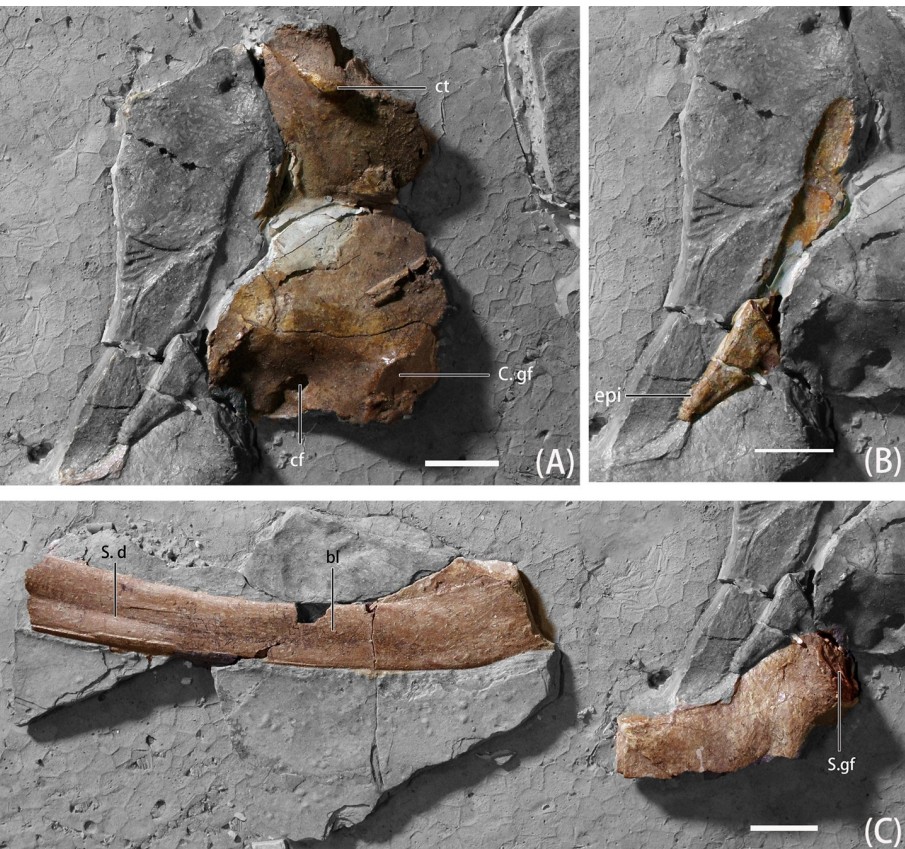

**Fig 5.** Photographs of pectoral girdle elements of *B. inexpectus* (IVPP V 11559) A. coracoids; B. furcula; C. scapula.
**Abbreviations: bl**, blade; **cf**, coracoid foramen; **C. gf**, glenoid fossa of coracoid; **ct**, coracoid tubercle; **epi**, epicleidium;
**S. gf**, Glenoid fossa of scapula; **S. d**, depression of scapula. Scale bars equal 2 cm.

a feature also seen in most other therizinosaurians and some maniraptoran theropods [8],
except the coracoids of *Jianchangosaurus*, which are more similar to those of ornithomimo-
saurs in having a semicircular shape [20]. The anterior region is anteroposteriorly shortened
and elongated dorsoventrally as in *Falcarius* [2], and unlike *Neimongosaurus*, in which the
anterior portion is plate-like and deflected posteromedially [23]. The curvature is relatively
continuous as in *Falcarius* [2], although not as strongly recurved, and unlike the hinge form in
*Neimongosaurus* [23]. There is a pronounced, nearly crest-shaped coracoid tubercle [8] similar
in form to *Falcarius* [19], that is best-preserved on the left coracoid, unlike the moderately
developed coracoid tubercle in *Alxasaurus* [6]. As in *Alxasaurus* and *Falcarius*, the coracoid
tubercle occurs close to the ventral margin of the glenoid [2, 6]. The coracoid comprises
slightly more than half of the glenoid. This condition is intermediate between that of the early-
branching *Falcarius* [2], in which the scapular portion is greater, and *Neimongosaurus*, in
which the coracoid comprises about two-thirds of the glenoid [23]. Dorsal to the glenoid, there
is a coracoid foramen, similar to that of *Falcarius* morphologically [2].

**Scapula.** A partial right scapula is preserved in medial view. There is damage to the proxi-
mal and distal ends (Fig 5C). The scapula and coracoid are not fused, which is a feature present
in early-branching therizinosaurians, including *Falcarius* [2], *Jianchangosaurus* [20], *Marthar-
aptor* [29], as well as some later-branching taxa, such as *Nothronychus* [27] and *Erliansaurus*
[32]. In other late-branching therizinosaurians, including *Suzhousaurus* [30], *Neimongosaurus*

**Table 2. Measurements in millimeters of pectoral girdle and forelimb elements of *B. inexpectus* (IVPP V 11559).**

| Element | Length | Width | Note |
|---|---|---|---|
| Scapula, right | >194.5* | 21.2** | *Losing distalmost part and some part of blade in proximal end is broken |
| | | | **Narrowest part |
| Coracoid | - | - | |
| Furcula | 84.8 | - | |
| Humerus, right | >163.9* | 47.0** | *Distal end is broken and some midshaft is losing. |
| | | | **Width is measured from humeral head to lateral side |
| Ulna, right | - | 13.4 | |
| Radius, right | >120.5* | 19.2 | *Losing some midshaft |
| Distal carpal II | 5.8 | 20.0 | |
| Distal carpal III, left | 15.8 | 15.0 | |
| MCII, right | 38.9 | 16.1 | |
| MCIII, right | 77.3 | 13.7 | |
| MCIV, right | 75.9 | - | |
| PII-1, right | 63.6 | | |
| PII-2, right | 71.7 | | |
| PIII-1, right | 50.9 | | |
| PIII-2, right | 60.1 | | |
| PIII-3, right | 86.1 | | |
| PIV-1 | - | | |
| PIV-2 | - | | |
| PIV-3, right | 38.5 | | |
| PIV-4 | - | | |

[23], *Segnosaurus* [4], and *Therizinosaurus*, the scapula and coracoid are fused. However, since IVPP V 11559 is a skeletally immature individual, the unfused condition might be the result of early ontogenetic stage. The blade is long and slender, with a slight distal expansion. Expansion of the distal blade can also be observed in *Falcarius* [2], *Alxasaurus* [6], *Neimongosaurus* [23], and *Erliansaurus* [32]. The reverse condition, scapula blade tapering of the distal end, is noted for later-branching therizinosaurians, such as *Nothronychus* and *Segnosaurus* [2, 4, 26].

The cross-section of the proximal end of the blade is flat and thin, gradually decreasing in dorsoventral height. There is a depression that begins midway along the medial surface of the scapular blade on *B. inexpectus*. This feature may be enhanced by or entirely the result of crushing. Such a depression is absent on *Falcarius*, *Neimongosaurus*, and *Erliansaurus*. The cross-section of the proximal blade is subtriangular, as in *Falcarius* [2]; with the transversely wider ventral base formed by a ridge in the medial surface. The glenoid fossa is a kidney-shaped facet that spans both the scapula and coracoid. It is oriented posteroventrally, similar to the condition found in early-branching therizinosaurians such as *Falcarius* [2] and *Jianchangosaurus* [20], and unlike the glenoid of the relatively late-branching *Neimongosaurus*, and *Nothronychus*, which faces more laterally [23, 26].

## Forelimb

Both forelimbs are preserved in IVPP V11559 and are relatively complete (Figs 6 and 7 and Table 2). Forelimbs are relatively longer than the short and stout hind limbs in *B. inexpectus*, a feature shared with many dromaeosaurids and early-branching avialans [8, 34]. This feature is also present in some other early-branching to intermediate therizinosaurians. For instance, the ratio of humerus to femur in other therizinosaurian dinosaurs is as follows: *Jianchangosaurus*

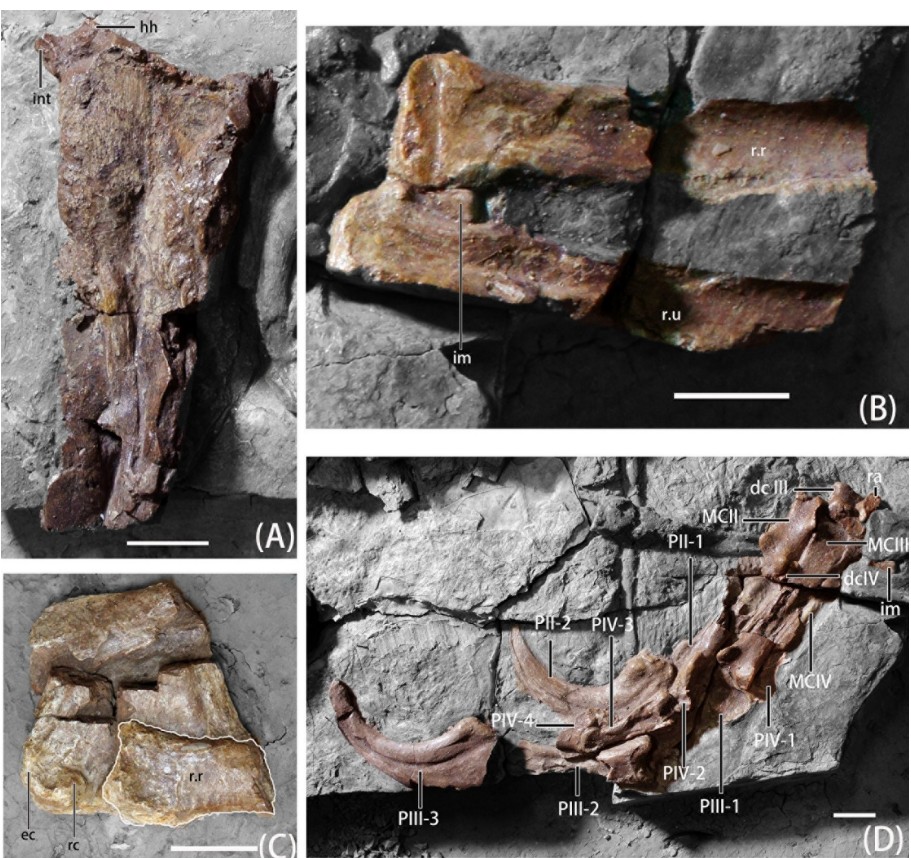

**Fig 6.** Photographs of right forelimb elements of *B. inexpectus* (IVPP V 11559) A. humerus in anterior view; B. radius and ulna; C. humerus distal end with radius; D. carpus and manus elements. **Abbreviations: dc III-IV**, distal carpal III-IV; **ec**, ectepicondyle; **hh**, humeral head; **im**, intermedium; **int**, internal tuberosity; **MCII-IV**, metacarpus II-IV; **PII-1 to PIV-4**, phalanges II-1 to IV-4; **ra**, radiale; **r.r**, right radius; **r.u**, right ulna; **rc**, radial condyles. Scale bars equal 2 cm.

76.7%, *Alxasaurus* 67.6%, *Erlianosaurus* 66.7%, *Neimongosaurus* 60.7%, and *Nothronychus* 38.7% [6, 20, 23, 27, 32]. Although the humerus of *B. inexpectus* is incompletely preserved, it is at least more than 60% of the femur length (Tables 2 and 3).

**Humerus.** Both humeri are preserved. The right one is complete and preserved in anterior view; however, its proximal and distal ends are separated (Fig 6A and 6B). Only the distal part of the left humerus, preserved in posterior view, remains (Fig 7B). The humerus is shorter than the scapula, and relatively straight. In early-branching therizinosaurians, (e.g., *Falcarius)*, the humerus is slightly sigmoidal, more similar to that of oviraptorids and *Deinonychus* [2]. Whereas, in late-branching therizinosaurians, the humerus varies from relatively straight [5, 26] to strongly sigmoidal [2, 4].

There is a pointed internal tuberosity in the proximal end of the humerus, with a depression to separate the tuberosity and humeral head, which is also present in most therizinosaurians and *Mononykus* [8, 35]. This depression extends across the posterior aspect to the proximal surface [8], as in *Neimongosaurus*, *Segnosaurus* and *Erlikosaurus*, [2]. However, we note that poor preservation of the humerus suggests caution in interpreting the morphology of the internal tuberosity, humeral head, and depression separating these features. In most therizinosaurians, the humeral head is hypertrophied both posteriorly and dorsally, appearing bulbous in form [2], whereas in IVPP V11559, only the dorsal extension of the humeral head can only be

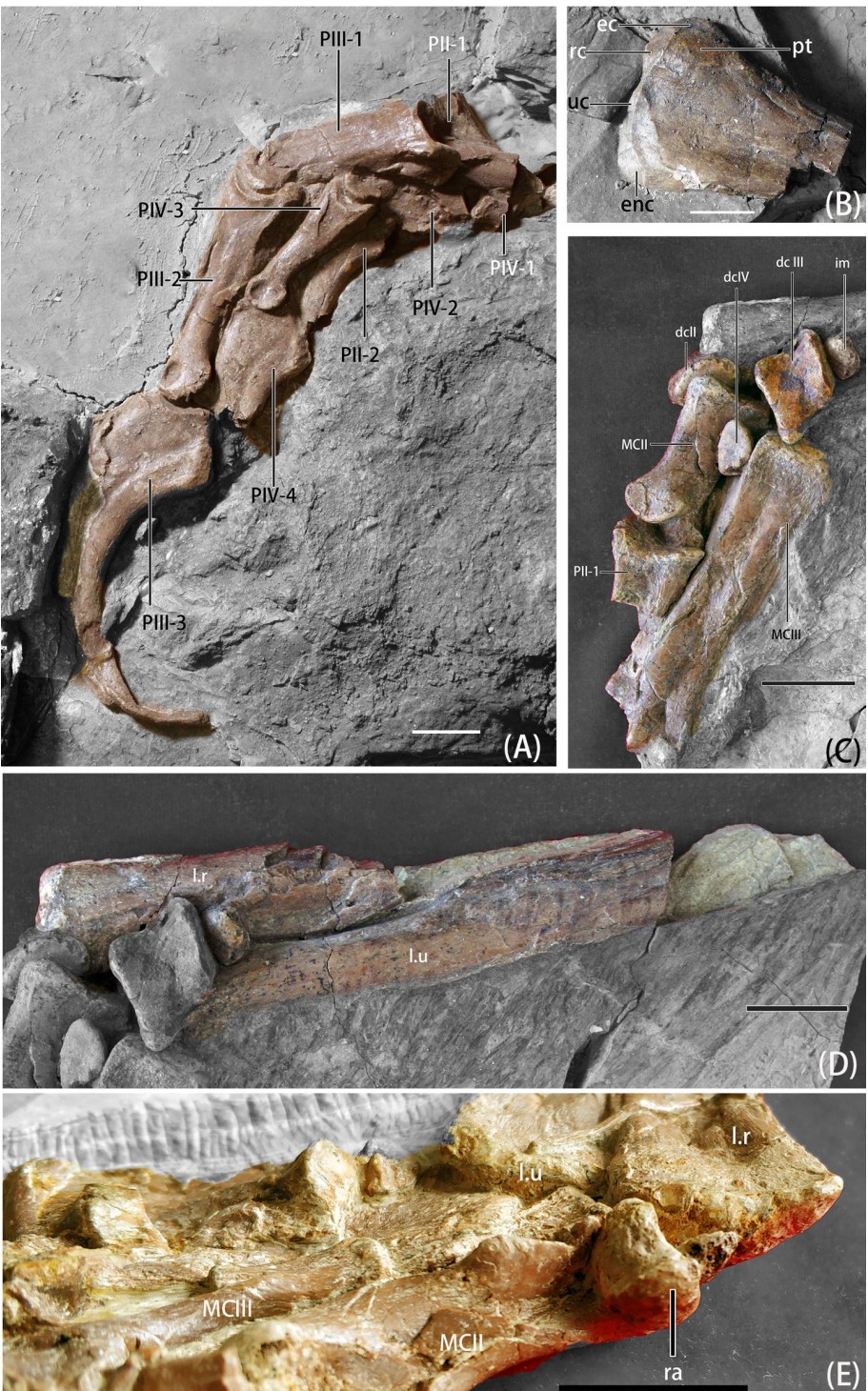

**Fig 7.** Photographs of left forelimb elements of *B. inexpectus* (IVPP V 11559) A. manus elements in lateral view; B. humerus distal end in posterior view; C. carpus and metacarpus; D. radius and ulna; E. manus elements in anterior view. **Abbreviations: ec**, ectepicondyle; **dcII-IV**, distal carpal II-IV; **enc**, entepicondyle; **l.r**, left radius; **l.u**, left ulna; **im**, intermedium; **MCII-III**, metacarpus II-III; **PII-1 to PIV-4**, phalanges II-1 to IV-4; **pt**, posterior tuberosity; **ra**, radiale; **rc**, radial condyles; **uc**, ulnar condyles. Scale bars equal 2 cm.

**Table 3. Measurements in millimeters of pelvic girdle and hindlimb elements of *B. inexpectus* (IVPP V 11559).**

| Element | Total length | Hight | Note |
|---|---|---|---|
| Ilium, right | >115.8 | 80.1* | *From acetabulum |
| Pubis | >233.3* | | *Losing distalmost boot part |
| Ischium | >150.0* | | *Losing proximal end |
| | **Length** | **Width, proximal** | **Width, distal** |
| Femur, right | 275.8 | 92.1 | 56.1 |
| Tibia, right | 271.2 | - | 65.7 |
| | **Length** | **Width** | |
| Astragalus | - | 56.7 | |
| Calcaneum | 32.5 | 20.3 | |
| MTI | 31.4 | 12.6 | |
| MTII | 96.5 | 25.3 | |
| MTIII | 105.7 | 26.3 | |
| ?PI-1 | 36.3 | | |
| Pedal ungual | 51.7 | | |

confirmed due to the poor preservation. Only the base of the deltopectoral crest is preserved. It extended from the humeral shaft 90 degrees. By observing the broken scar, the deltopectoral crest should be about one-third of the humeral length, similar to the ratio of *Falcarius*, *Jianchangosaurus*, *Neimongosaurus*, and *Erliansaurus* [2, 20, 23, 32], and shorter in form than in *Segnosaurus* and *Therizinosaurus* [4, 5].

As in other early-branching therizinosaurians, the proximal and distal ends of humerus are moderately expanded, narrowest in mid-shaft [20]. The ulnar and radial condyles are positioned anteriorly as in most therizinosaurians [2, 8]. Close to the lateral surface of distal ends, there is a significant posterior tuberosity opposite the radial condyle. Next to the tuberosity and extending laterally is a dorsoventrally elongate ectepicondyle. The entepicondyle is hypertrophied and extends anterolaterally. On the posterior surface of the distal end, there is a ridge originating from the hypertrophied entepicondyle. Concordant with this ridge, there is a groove on the posterior surface closer to the medial side, which separates the condyles of the distal end.

**Ulna.** Both ulnae are partial and preserve the distal aspect. The left ulna is more complete including the posterior surface (Figs 6B and 7D). Although a proximal end of the right ulna was noted as being present by Xu, et al. [8], the element could not be found at the IVPP during this study. Based on the preserved portions, the ulna is estimated to be shorter than the humerus, as in most therizinosaurians. The ratio of the ulna to the humerus in early-branching therizinosaurians such as *Falcarius* and *Jianchangosaurus* is 77–78% [2, 20], and *B. inexpectus* might be close to this ratio. Among late-branching therizinosaurians the ratio of these elements ranges from 72% in *Nothronychus* [26] to as elongate as 85% in *Erliansaurus* [32]. The preserved parts and the impression show that the mid-shaft of the ulna is straight and not bowed, like in most other therizinosaurians such as *Jianchangosaurus*, *Nothronychus*, *Erliansaurus*, and *Therizinosaurus* [20, 26, 32], but different from that of *Falcarius*, which is bowed [2]. The cross-section of the ulnar shaft of *Beipiaosaurus* is oval medially, becoming flattened near the distal end and is semicircular in distal view. There is a prominent, medially projecting tubercle on the distal end, which is anteroposteriorly flattened, similar to *Falcarius* [2].

**Radius.** Both distal radii are preserved, along with the proximal end of the right radius (Figs 6B and 6C and 7D). Although the diameter of the radius is larger than the ulna, it is still gracile in shape. A gracile radius is also present in *Falcarius*, *Jianchangosaurus*, and *Neimongosaurus* [2, 20, 23]. The radii of later-branching therizinosaurians are more massive and robust

[4, 5]. Based on the preserved parts and the impression, the radius of *B. inexpectus* is a relatively straight element, like *Falcarius* and *Jianchangosaurus* [2, 20]; whereas in *Therizinosaurus* and *Neimongosaurus*, this element is relatively sigmoidal [2, 4, 23]. The proximal end of the radius is oval and medially deflected to form the proximal fossa. The cross-section of the shaft is subcircular, becoming subtriangular and mediolaterally expanded approaching the distal end, which flattens medially to articulate with the distal ulna.

**Carpus.** Nine elements of the carpus are preserved. Present are distal carpals II, III, and IV, the intermedium, and the radiale of the left wrist, and distal carpal III and IV, the intermedium, and the radiale of the right wrist (Figs 6D and 7C and 7E).

The radiale of the left manus is V-shaped, with a depression on the proximal end, near the contact with the radius (Fig 7E; [8]). Positionally, the right radiale is also preserved near distal carpal III, and has a rod-like shape (Fig 6D), but the poor preservation hinders further identification and description. Both intermedia are preserved. Considering both of the elements, the intermedium of *B. inexpectus* is a flat and round element with a wedge-shaped, narrower end. Intermedia are either fragmented or not exposed well, so the articulation with the other carpals, and the precise morphology, are difficult to identify.

There are morphological and positional discrepancies in the preserved carpals that prevent us from confident identification of these elements. We present both hypotheses here.

If we identify the distal carpals by their preserved position relative to the metacarpals, the distal carpal II of *B. inexpectus* is large and oval [8], and it is unfused to distal carpal III. Whether this is related to ontogenetic stage is unclear. In *Falcarius*, distal carpals II and III are partially fused dorsally, and together these elements make up the semilunate [2]. The distal carpals of *B. inexpectus* differ from those of *Alxasaurus*, in which the distal carpal II is the largest [6]. In *B. inexpectus*, distal carpal III is larger than distal carpal II. Distal carpal III is slightly elongate and caps primarily the proximal surface of metacarpal III, and touches the proximal surface of metacarpal II [8]. In proximal view, distal carpal III is convex dorsally and ventrally, with a well-defined trochlear groove for the radiale. Although the semilunate (distal carpal III) is smaller and the anterior process is blunter and more reduced, the overall shape of it is identical to that of *Deinonychus* [8, 36]. In early-branching therizinosaurians such as *Falcarius*, the fused distal carpal II and III is larger and more robust than other therizinosaurians and dromaeosaurs [2].

However, we recognize a different possibility for the identifications of distal carpal II and III. Morphologically, it seems that the carpal element termed the "semilunate" in Xu et al. [8] and also labelled as "dc III" on our intext figures is most similar to the single carpal distal carpal II in *Falcarius*, and that the "semilunate" carpal structure would be comprised of both distal carpal II and distal carpal III in early branching therizinosaurs [2, 37]. Also, based on the size ratio of the distal carpals and the morphology of the "semilunate" (distal carpal III) in *B. inexpectus*, this element is more identical to that of the distal carpal II of *Alxasaurus* [6, 37]. In this case, it is highly possible that the positionally consistent distal carpal II and III are translocated during the preservation or the preparation, and they are actually distal carpals III and II, respectively.

The distal carpals IV are also incomplete on both manus. The distal carpus IV is a proximodistally thin element, and tapers on the lateral end giving it a subtriangular shape. The articular facet is slightly concave. Distal carpal IV is missing in other therizinosaurian taxa, so there is little else to support the identification of this element.

## Manus

The manus of IVPP V11559 is relatively complete, and missing parts can be inferred from impressions (Figs 6D and 7A). In general, the manus of *B. inexpectus* is elongate and slender, with well-developed ginglymoid articulations and raptorial unguals. Digit IV is shorter than digit III.

**Metacarpus.**   All the metacarpals of *B. inexpectus* are preserved, except for the left metacarpal IV, (Figs 6D and 7A). In general, metacarpal elements of *B. inexpectus* resembles to those of *Falcarius* [2].

Metacarpal II is the most robust and shortest of the metacarpals, only half the length of metacarpal III. The cross-section is subtriangular with a laterally flattened facet for articulation with metacarpal III. There is a concavity on the proximal surface of metacarpal II that extends to the dorsal surface between the medial process and the dorsal process, forming a notch. There are three processes on the proximal articular surface, which make a triradiate shape. This condition is also present in the early-branching therizinosaurian *Falcarius* [2]. Among the three processes, the medial process extends farthest to support the dorsomedial extension of distal carpal I. The same condition can also be seen in *Falcarius* and *Jianchangosaurus* [2, 20], but not as extremely as in *Erliansaurus* [32]. The lateroventral surface of the proximal end of metacarpal II is pointed and extends to underlie the base of metacarpal III; this is known as the "rectangular buttress". This character was first described in *Alxasaurus* and as a synapomorphy for Therizinosauroidea [6]. Although this buttress is also present in the earlier-branching *Falcarius*, it is more moderate [2]. The distal end of metacarpal II is slightly broader and asymmetrical, with a more extensive lateral condyle and a medially rotated axis. This feature is also seen in *Falcarius*, *Jichangosaurus*, and *Erliansarurus* [2, 20, 32]. As in *Falcarius* and *Therizinosaurus*, the distal end of metacarpal II lacks collateral ligament pits and a dorsal extensor pit.

Metacarpal III is the longest of the metacarpals, and has transversely expanded and robust ends. The shaft is rectangular proximally and becomes oval and dorsoventrally flattened in the midshaft. The proximal articular surface is rectangular and convex, unlike the proximal articulation surface of metacarpal II, which is concave. Each margin of the rectangular proximal surface is divided by rounded tuberosities, and a dorsolateral tuberosity extends to overhang the proximal end of metacarpal IV. This morphology is similar to that of *Falcarius* [2]. The distal end of metacarpal III is asymmetrical, with a vertically oriented, rounded lateral condyle and a larger medial condyle that is angled outward. A larger medial condyle is also described in *Falcarius* and *Erliansaurus*, but in *Falcarius* it is less prominent [2, 32].

Metacarpal IV is only preserved on the right manus. By observing the impression, metacarpal IV can be inferred to be a thin, long, and straight element, more gracile than metacarpal II and metacarpal III. This character is similar to that of *Faclarius* morphologically [2]. The length of metacarpal IV is about 75% of Metacarpal III. The shaft of metacarpal IV is oval in cross-section. The proximal end of metacarpal IV is convex and subtriangular; whereas the distal end is slightly bowed toward metacarpal III.

**Manual phalanges.**   The left arm of *B. inexpectus* is more complete than the right one. There are many missing and incomplete right phalanges, but clear impressions of these missing elements remain (Figs 6D and 7A). There are well-developed ligament pits on the lateral sides of distal ends. All the manual unguals are deep, compressed mediolaterally, and curved strongly [6, 8]. As in *Falcarius*, the collateral grooves of the manual unguals are not continuous with large depressions in the medial and lateral surface of the proximal aspect [2]. In *Lingyuanosaurus* and *Jianchangosaurus* these features connect [21]. Collateral grooves rise dorsally toward the distalmost tip, as in many other therizinosauroids, including *Lingyuanosaurus*, *Alxasaurus*, and *Therizinosaurus* [6, 21]. Manual unguals are subequal in length with the penultimate phalanges in *B. inexpectus*. Unlike, in the therizinosaurids *Nothronychus* and *Therizinosaurus*, in which the unguals are longer than the penultimate phalanges ([2], Fig 5; [31], Fig 1).

Digit II is the shortest of the three digits, but phalanx II-1 is the longest and the most robust phalanx. The same condition can also be seen in *Jichangosaurus* and *Falcarius* [2, 20]. The proximal articular surface of phalanx II-1 is subtriangular and taller than wide. Although the

midshaft is missing, from the impression and the preserved distal shaft, it can be inferred that the midshaft is straight with a subcircular cross-section. The distal ginglymoid articular surface is symmetrical with a deep intercondylar groove, and the condyles are more extensive ventrally than dorsally. Collateral ligament pits are well developed and positioned more dorsally. The manual ungual of digit II of most therizinosaurians is the longest and largest including *Jianchangosaurus* [20], *Alxasaurus* [6], and *Erliansaurus* [32]. However, the manual ungual II of *B. inexpectus* is slightly shorter than manual ungual III, and it possesses the strongest degree of curvature. As in other early-branching therizinosaurians, *B. inexpectus* also possesses a deeper proximal articular surface and more massive flexor tubercle similar to that of oviraptorosaurians [2, 6, 32]. The flexor tubercle and the ventral margin of the proximal articular surface of phalanx II-2 are separated by a narrow and deep groove.

Digit III is the longest digit, phalanx III-1 is about 80% length of phalanx II-1, and phalanx III-2 is only slightly shorter than phalanx II-1. Phalanx III-1 is stout in shape, with a hexagonal proximal surface. In proximal view, the dorsal intercondylar tuberosity is less pronounced, but the ventral intercondylar tuberosity is relatively robust. On the distal end, collateral ligament pits are poorly developed and the pits are positioned nearly centered on the condyle. In phalanx III-2, the cross-section of the proximal end is oval, and dorsoventrally taller than wide. The shaft of phalanx III-2 of *B. inexpectus* is straight, and gradually tapers in dorsoventral height toward the distal end. Nonetheless, it remains taller than wide in cross-section. On the distal end of phalanx III-2, the intercondylar groove is deep, the condyles are taller than the height of the mid-shaft, and the collateral ligament pits are located more dorsally. These features of the distal end are characteristic of penultimate phalanges [2]. Manual ungual III is less re-curved than manual ungual II, but longer. A longer second claw is a character similar to that of *Archaeopteryx* and *Protarchaeopteryx* [8, 38]. The proximal articular surface and flexor tubercle of phalanx I-3 are divided by a broad and shallow notch. The flexor tubercle of manual ungual III is narrower and smoother than that of manual ungual II, and lacks the distal pits present in manual ungual II. On the tip of the left ungual III, there is a dorsoventrally flat extension that might represent preservation of the keratin sheath. If so, the claw sheath of *B. inexpectus* seems to be more flattened, rather than a sharply pointed structure, but this might be caused by taphonomic processes.

Digit IV is relatively gracile, with extremely shortened phalanges IV-1 and phalanx IV-2. When combined, the length of these two phalanges is equal to that of phalanx IV-3 [8]. Digit IV is in poor condition, most parts of phalanx IV-1 and phalanx IV-2 are missing, so it is difficult to observe any detailed structure. Phalanx IV-3 is relatively complete. The shape of phalanx IV-3 is gracile and elongate, with a mediolaterally narrow and oval cross-section. The collateral ligament pits of the distal end are well developed and located dorsally. Phalanx IV-4 is the smallest manual ungual of the three digits. The flexor tubercle, the groove that separates it from proximal joint surface, as well as the lateral sulci of phalanx IV-4, are the most poorly developed of any manual ungual.

## Pelvic girdle

Pelvic girdle elements of IVPP V11559 reported in 1999 [8] include a partial ilium, pubis and ischium. Materials found during the re-excavation of the same quarry and reported in 2003 include a complete ilium and two incomplete ischia (Fig 8 and Table 3) [11]. *B. inexpectus* processes features such as a parallelogram-shaped ilium and propubic pelvis that are more primitive than the morphology of late-branching therizinosaurids [3, 8].

**Ilium.**    The left ilium of *B. inexpectus* is partially preserved and exposed in medial view (Fig 8A and 8B). The right ilium was found in the re-excavation. It is relatively complete and

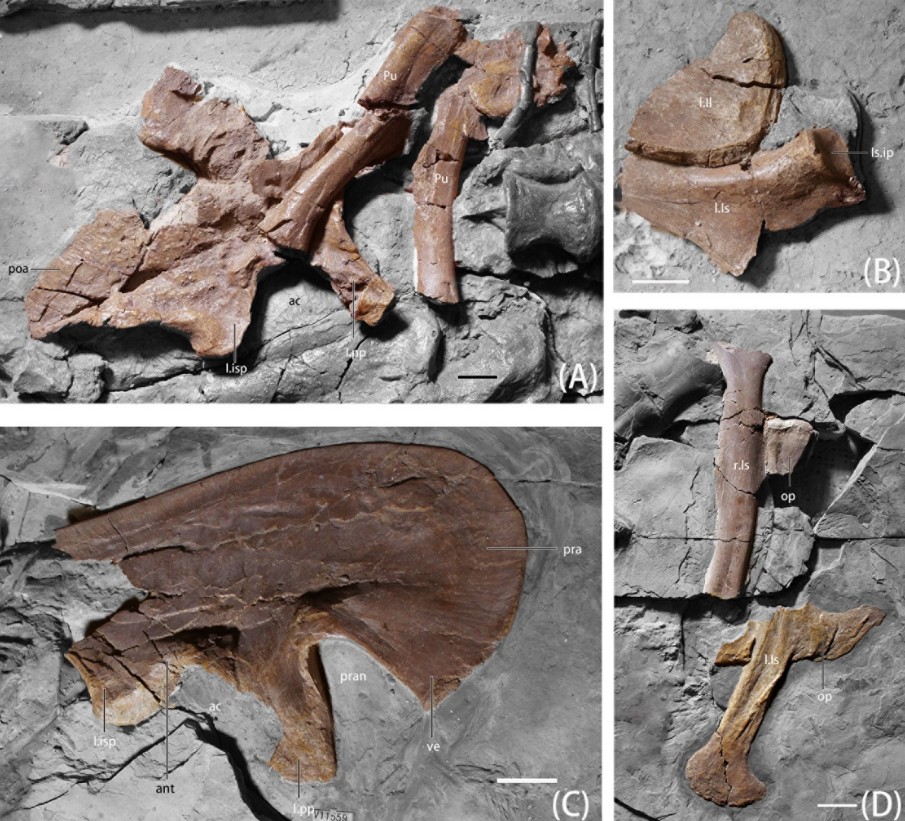

**Fig 8.** Photographs of pelvic girdle elements of *B. inexpectus* (IVPP V 11559) A. left ilium in medial view with pubis; B. partial ilium and ischium; C. right ilium in lateral view; D. right ischium in lateral view and left ischium in medial view. **Abbreviations: ac**, acetabulum; **ant**, antitrochanter; **I.isp**, ischiadic peduncle of ilium; **I.pp**, pubic peduncle of ilium; **Is.ip**, iliac peduncle of ischium; **l.Il**, left Ilium; **l.Is**, left ischium; **op**, obturator process; **poa**, postacetabular; **pra**, preacetabular; **pran**, preacetabular notch; **Pu**, pubis; **r.Is**, right ischium; **ve**, ventral extension. Scale bars equal 2 cm.

exposed in lateral view (Fig 7C). The ilium is parallelogram-like in shape, similar to that of dromaeosaurids and early-branching birds, but different from that of late-branching therizinosaurians, which have an ilium more similar in shape to sauropods [8, 39]. The length of the pre- and postacetabular portions of the ilium are subequal, as in many oviraptorosaurians and early-branching ornithomimosaurians [3, 8, 40, 41]. The relative proportion of the pre- and postacetabular aspects of the ilium is difficult to quantify in other therizinosaurians except for some taxa, such as *Falcarius*, for which the preacetabular portion is approximately 30% longer than the postacetabular portion. *Nothronychus* and *Segnosaurus* also possess longer preacetabular portions, but the ratio is not as extreme as in *Falcarius* [19, 31].

The preacetabular blades of therizinosaurians are dorsoventrally deep, dubbed the "altiliac" condition, and bear a pointed ventral extension. This extension is extreme in *Lingyuanosaurus*, *Segnosaurus*, and *Nanshiungosaurus*, extending ventral to the dorsal margin of acetabulum [3, 21, 42]. However, this character is less significant in *B. inexpectus*, in which case the ventral aspect of the preacetabular portion of the ilium only extends slightly ventral to the dorsal acetabulum [11]. Early-branching therizinosaurians such as *Falcarius* and *Jianchangosaurus* lack this feature [19, 20]. The preacetabular notch is considerably higher than the acetabulum as in other therizinosaurians including the early-branching *Falcarius*, but in theropods, the dorsal margin of the acetabulum is subhorizontal with the ventral aspect of the preacetabulum [19].

The supracetabular crest is relatively mediolaterally deep, and entirely capped the femoral head. The dorsoventral height of the postacetabular portion gradually reduces posteriorly in lateral view. It is subrectangular in shape, with a blunt posteroventral corner. This feature is already present in the earliest-branching therizinosaur *Falcarius*, and is a character of all therizinosaurians. The posteroventral aspect of the ilium is deflected laterally at a right angle to the vertical ramus. The brevis fossa is shallow, subcircular and oriented mediolaterally, which is similar to coelurosaurians generally [8, 43].

The pubic peduncle of *B. inexpectus* is longer than the ischiadic peduncle, as in therizinosaurians, dromaeosaurids, and *Archaeopteryx* [8, 43], but the length is subequal in earlier-branching therizinosaurians (e.g., *Falcarius* [19]). The pubic peduncle is equidimensional, unlike that of earlier-branching *Falcarius* and *Jianchangosaurus*, in which the anteroposterior length is as twice that of the mediolateral width; whereas in later-branching therizinosaurians, the mediolateral width is two or three times longer than the anteroposterior length ([19, 20], Fig 9). *Beipiaosaurus inexpectus* is intermediate between these values. The middle of the pubic peduncle is posteriorly concave, which is also seen in other late-branching therizinosauroids, including *Lingyuanosaurus*, *Suzhousaurus*, and *Nanshiungosaurus* [21, 22, 30]. The articular facet of the pubic peduncle is oriented anteroventrally, which when combined with the morphology of the proximal pubis creates an apropubic pelvis. The ischiadic peduncle is peg-like. These characters are similar to *Falcarius* and *Jianchangosaurus* [19, 20]. There is a prominent antitrochanter on the dorsolateral surface of the acetabulum, which rounds out the acetabulum in lateral view.

**Pubis.**   Both pubes are preserved, lacking proximal and distal ends (Fig 8A). There is a slight curvature of the distal shaft creating a sigmoid shape. The shaft is anteroposteriorly compressed and drop-shaped in cross-section as in *Falcarius* [19], but different from that of other late-branching therizinosaurians including *Alxasaurus*, *Segnosaurus*, *Nothronychus*, and *Shuzhousaurus*, which have more transversely flattened shafts [8, 30, 31]. The pubic apron is compressed anteroposteriorly, and might have extended more than half the length of the shaft as in *Falcarius* [19], but due to the fact that both ends are missing, the proportions cannot be confidently calculated. The pubic boot in *B. inexpectus* is yet unknown. The ischial boot from the 2003 specimen was erroneously compared to the pubic boot of *Falcarius* in Zanno [19]; this error was subsequently corrected in Zanno [3].

**Ischium.**   Two incomplete ischia are present, preserving the proximal portion of the right ischium and the shaft and distal portions of both the right and left ischia ([11], Fig 8D; [8], Fig 8B). The ischium is estimated to be subequal or slightly shorter than the pubis. The relative length of the ischium is shortest in *Falcarius*, which is only about 2/3 the length of the pubis; in *Jianchangosaurus*, the ischium is about 20% shorter; and in late-branching therizinosaurians, the length of these two elements are subequal [19, 20].

Ischia are slightly curved posteriorly and mediolaterally compressed. The lateral surface is slightly convex and the medial surface is flattened. The proximal portion is damaged, but bears a fan-shaped expansion in lateral view, and might be similar to *Faclarius*, which possesses a longer iliac peduncle than pubic peduncle [19].

On the anterior surface of the mid-shaft, closer to the proximal end, there is a prominent, dorsoventrally expanded obturator process. It is different from the condition of late-branching therizinosaurians, which have a hypertrophied and distally positioned obturator process [19, 20]. The distal point of the obturator process in *B. inexpectus* first projects anterodorsally and then anteroventrally, forming a sinusoidal process, unlike in *Falcarius* and *Jianchangosaurus*, in which it points anteriorly [19, 20]. The position of obturator process in *B. inexpectus* and *Jianchangosaurus* is relatively close to the mid-shaft, which is similar to that of tyrannosaurs and early-branching ornithomimosaurs [41, 44]. On *Falcarius*, the obturator process is

positioned more distally, as in dromaeosaurs, troodontids, early-branching birds, and oviraptorosaurs [18, 19, 36, 40, 45–50].

The distal end of the ischium bears a prominent, anteriorly expanded, and semicircular boot, in medial view. The anterior margin is relatively pointed and convex, and the posterior margin is round and blunt. A prominent distal ischial boot is different from that observed on *Jianchangosarus*, *Alxasaurus*, and *Nothronychus*, in which the boot is only expanded or thickened slightly [6, 20, 27].

## Hindlimb

Only the right leg is preserved on IVPP V11559; it is relatively complete (Fig 9 and Table 3). Although Xu, Tang, and Wang [8] describe the tibia of IVPP V 11559 as being longer than the femur, our measurement shows that they are subequal in length and the femur is slightly longer (Table 3). The hindlimb of *B. inexpectus* is intermediate in morphology, including the relative ratios of the hind limb elements. The ratio of femur/tibia and metatarsus/tibia in therizinosaurians is as follows: *Falcarius* 91/45%, *Jianchangosaurus* 65/54%, *B. inexpectus* 101/39%, *Nothronychus graffmi* 104/35%, *Neimongosaurus* 118/37%, *Erliansaurus* 110%/unknown, and *Segnosaurus* 102/32% [8, 19, 20]. The ratio in *B. inexpectus* fits its phylogenetic position between earliest-branching taxa (*Falcarius* and *Jianchangosaurus*) and therizinosaurids.

**Femur.**   The right femur is preserved and exposed in anterior view (Fig 9A). Both *B. inexpectus* and *Falcarius* possess relatively gracile femora that display a mosaic of late-branching maniraptoran and early-branching coelurosaurian features [19]. The femur of *B. inexpectus* is somewhat intermediate between the recurved condition of *Falcarius* and the early-branching

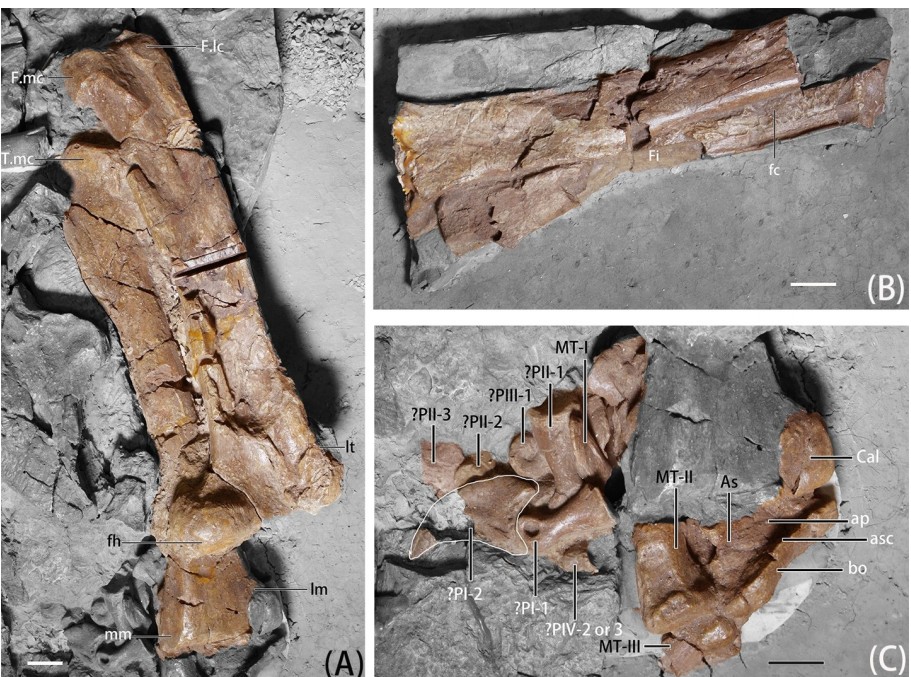

**Fig 9.**  Photographs of hindlimb elements of *B. inexpectus* (IVPP V 11559) A. right femur in anterior view and tibia in posterior view; B. left tibia and fibula; C. right tarsus and pedal elements. **Abbreviations:** ap, ascending process; **As**, astragalus; **asc**, astragalus condyle; **bo**, body; **Cal**, calcaneum; **Dt**, distal tarsal; **F. lc**, lateral condyle of femur; **F. mc**, medial condyle of femur; **fc**, fibular crest; **fh**, femoral head; **Fi**, fibula; **lm**, lateral malleolus; **lt**, lesser trochanter; **mct**, medial crest; **mm**, medial malleolus; **MT I-III**, metatarsus I-III; **?PI-1 to ?PIV-3**, ?phalanges I-1 to IV-3; **T.mc**, medial condyle of tibia. Scale bars equal 2 cm.

straight femur of many other therizinosauroids, including *Jianchangosaurus*, *Lingyuanosaurus*, *Erliansaurus*, and *Neimongosaurus* [20, 21, 23, 32].

The femoral head of *B. inexpectus* in anterior view is perpendicular to the long axis of the shaft, similar to *Falcarius* and *Jianchangosaurus* [19, 20]. This feature is different from that of late-branching therizinosaurians, such as *Lingyuanosaurus*, *Alxasaurus*, and *Nothronychus*, which possess a dorsomedially rotated femoral head [6, 21, 31]. The femoral head is large and bears a distinctly rounded head and weakly constricted neck, similar to the condition in *Jianchangosaurus*, but different than in *Falcarius* and *Lingyuanosaurus*, which have a more distinct and constricted femoral neck [19, 21]. As in *Falcarius* and *Jianchgosaurus*, the femoral head of *B. inexpectus* is continuous with the greater trochanter. This is unlike the condition in late-branching therizinosaurians such as *Lingyuanosaurus*, *Nothronychus*, and *Neimongosaurus*, which have a depression between the femoral head and the greater trochanter in anterior view [19–21]. This depression is not only present in late-branching therizinosaurians, but also in oviraptorosaurians and alvarezsaurs [19, 40, 51]. However, the absence of this feature in early-branching therizinosaurians indicates that it is not synapomorphic for these clades.

The lesser trochanter of *B. inexpectus* is wing-like and separated from the greater trochanter by a narrow and deep cleft, which is a character present in early-branching coelurosaurs [8, 19]. Both *Falcarius* and *B. inexpectus* possess an alariform lesser trochanter, but the shape is cylindrical in *Alxasaurus* and *Jianchangosaurus*, and even reduced in *Alxasaurus* [6, 19, 20]. The fourth trochanter projects from the posterior surface of the femur, with a ridge shape that is transversely thin and proximodistally long. The cross-section of the femoral shaft is subtriangular.

In anterior view there is a slight depression separating the medial condyle and the lateral condyle on the distal femur. In anterior and distal view, the distal condyles are asymmetrical, with a slightly larger and more extensive medial condyle than lateral condyle, which is opposite to that of *Falcarius*, which possesses a more extensive lateral condyle [19].

**Tibia.**   Both tibiae are preserved, and the right tibia is relatively complete (Fig 9A and 9B). In general, the shape of the tibia of *B. inexpectus* is gracile and slender, with some proportions that are more primitive than those of late-branching therizinosaurians.

The posterior surface of the right tibia of *B. inexpectus* is exposed and bears an asymmetrical proximal end in posterior view. On the proximal end, the medial condyle is dorsally higher and positioned more proximally, which is similar to that of *Falcarius* [19]. This asymmetrical proximal end in *Falcarius* corresponds with the less extensive medial condyle of the femur. However, this is not the case for *B. inexpectus* because the medial condyle of the femur is more distally extensive.

The tibia has a relatively straight shaft and its cross-section changes from subtrangular proximally to anteroposteriorly flattened distally. Due to the preservation position, the lateral side of the right tibia cannot be observed, but in the partial left tibia, there is a prominent fibular crest extending to the mid-shaft. Although it is damaged, it can be determined that it is more developed than in *Falcarius* [19], and less developed than in late-branching therizinosaurians such as *Nothronychus* and *Erliansaurus*, which have fibular crests that extend distally to the mid-shaft [31, 32].

The distal articular facet of the tibia of *B. inexpectus* is anteroposteriorly narrow and transversely broad, with a blunt and robust medial condyle, similar to *Falcarius*. But the medial condyle of *Falcarius* is orientated anteromedially [19], whereas in *B. inexpectus* it is orientated medially. The lateral condyle is large and semicircular in shape to support the fibula.

**Fibula.**   An incomplete left fibula is preserved, lacking the distal end (Fig 9B). In general, the fibula is much more slender and more gracile compared to tibia, especially the distal portion. The fibula of *B. inexpectus* lacks a medial fossa on the proximal portion, as in other

therizinosaurians, oviraptorosaurians, and dromaeosaurs [19, 26, 52]. Similar to *Alxasaurus* and some avialans, the medial surface of fibula is relatively flattened [6, 8, 53].

**Tarsus.** The right astragalus, calcaneum and a distal tarsal are preserved and exposed in posterior view (Fig 9C). They are unfused with the distal tibia, proximal metatarsals, and with each other. Lack of fusion may be the result of ontogeny, but in *Faclarius* these elements are also not fused even in the largest individual [19]. As in other therizinosaurians, the astragalar condyle is reduced, so it only capped a portion of the distal end of the tibia. In *Faclarius*, the astragalus and the calcaneum are relatively larger and cap the distal tibia and fibula entirely [5, 8, 19, 31].

The astragalus consists of an asymmetrical and robust, saddle-shape body, with a significantly larger medial surface. In posterior view, the medial aspect is thick proximodistally, and gradually narrows laterally. This condition is shared between *B. inexpectus* and *Falcarius* [19]. However, reduction of the astragalar body in late-branching therizinosaurians causes this feature to be absent in late-branching members [19]. As in other therizinosaurians, *B. inexpectus* also possesses a tall ascending process. The ascending process and the medial margin of the body is inset, with the base of the ascending process rising vertically.

The calcaneum is sub-oval and disk-shaped [8], with a slightly convex surface on the medial side. According to Xu et al. ([8], Fig 2e), a single distal tarsal is present, which attaches to the proximal surface of metatarsal IV. However, this element can no longer be located.

**Metatarsus.** Five elements of the right metatarsus are preserved, and in relatively complete condition (Fig 9C) [8]. However, due to the position of preservation, only portions of these elements are exposed and can be observed. As in *Falcarius* and *Jianchangosaurus*, *B. inexpectus* has a typical functionally tridactyl, and relatively gracile pes, in contrast to late-branching therizinosaurians, which have a stout and tetradactyl pes [8, 19, 20]. The metatarsus of *B. inexpectus* is compact proximally and elongate in general (~39% the length of the tibia). This ratio is longer than other late-branching therizinosaurians but shorter than *Falcarius* and *Jianchangosaurus*, as well as most theropods, for this ratio is typically over 45% [6, 8, 19, 20].

Metatarsal I of *B. inexpectus* is relatively robust, with a tapered and flattened proximal portion for articulation with metatarsal II. Metatarsal I articulates with metatarsal II on the mid-shaft and does not contact the tarsus, in contrast to the functionally tetradactyl pes of late-branching therizinosaurians.

In medial view, metatarsal II is slightly sigmoidal, with a proximal portion that is slightly curved anteriorly and a distal portion that curves posteriorly. In proximal view, the proximal articulation surface is mediolaterally narrow, oval in shape, and the lateral surface is flattened to correspond to the medial surface of metatarsal III. Metatarsal II is slightly narrower anteroposteriorly in the mid-shaft, especially the anterior surface, which slopes more than the posterior side, but it is still oval in shape (anteroposterior length is longer than the width) in general and expanded distally. The distal end of Metatarsal II of *B. inexpectus* is fan-shape in medial view and expanded distally, with a smooth medial surface, unlike *Falcarius*, which has a subcircular distal end and collateral ligament pits [19].

Metatarsal III and metatarsal IV are longer than metatarsal II. Metatarsal III is the slenderest and the longest of the metatarsals, and it is transversely compressed. According to Xu et al. ([8], Fig 2e), metatarsal IV and metatarsal V are preserved. However, metatarsal V and the proximal end of metatarsal IV can no longer be located.

**Pedal phalanges.** Pedal elements are disarticulated and partially broken, so it is difficult to identify them. Five pedal phalanges are preserved and exposed relatively well, and we tentatively identify them as phalanxes I-1, I-2, II-1, III-1, and IV-2 or 3 (Fig 9C).

A well-exposed phalanx near metatarsal I is tentatively identified as phalanx I-1. This phalanx is relatively robust, with a subcircular proximal articulation surface. The cross-section

becomes narrower from the proximal to the distal mid-shaft. Both the ginglymoid articular surface and extensor ligament pits are poorly developed on the distal end, but the collateral ligament pits on the sides are deep and well-developed and positioned centrally. The distal articular surface of this phalanx is relatively symmetrical. A single pedal ungual is preserved next to the referred phalanx I-1 that we interpret as phalanx I-2. It is short and curved, and smaller than any manual ungual [8].

Near the referred phalanx I-1, there is an elongate element interpreted as phalanx II-1, which possesses a subcircular proximal articular surface. Although it is poorly developed, an extensor ligament pit can be seen on the ventral side of the distal end. The shaft is curved laterally in ventral view. Articulating with the distal end of this phalanx, there is a shorter, more robust phalanx, likely to be phalanx II-2, and beyond this the proximal portion of phalanx II-3. Beneath the second digit rests another phalanx that we tentatively identify as phalanx III-1. This distal portion of this phalanx is exposed in lateral view and the collateral ligament pits of it are shallower than phalanx I-1.

## Discussion

It has been more than twenty years since *B. inexpectus* was first described. Since this time, multiple new early-branching and late-branching therizinosaurs have been discovered and described (e.g. *Falcarius* [7], *Jianchangosaurus* [20], *Lingyuanosaurus* [21], *Nothronychus* [26]). In addition, new materials of *B. inexpectus* and *B.* sp. have been published, including the recovery and description of additional materials of the holotype representing the pelvic girdle elements, sacrals, and a series of nearly complete caudals [11]. A new feather type was additionally described from around the tail of the holotype of *B. inexpectus* and the neck of a new specimen, *B.* sp. [12]. Based on these new discoveries, Zanno [19] revised the autapomorphies of *B. inexpectus*. This diagnosis was further revised after the cranial osteology was published by Liao and Xu [13] and several cranial diagnostic features were added. Although new taxa of therizinosauroids have since been described, the autapomorphies proposed by Zanno [19] and Liao and Xu [13] appear to remain valid.

### New acquired diagnostic features

In this study, we propos three new autapomorphies for *B. inexpectus*.

**Ungual II shorter than III.**   When measuring ungual length as a straight line from the tip of the ungual to the ventralmost portion of the articular facet, the ungual of digit three is the longest one in *B. inexpectus*. This feature differs from other therizinosaurians, including *Jianchangosaurus*, as well as later-branching *Alxasaurus* and *Erliansaurus* [2, 6, 20, 32], in which the ungual of digit two is the longest.

**The pre- and postacetabular length are subequal.**   In *B. inexpectus*, the pre- and postacetabular of ilium are subequal in anteroposterior length. This feature is similar in oviraptorids and early-branching ornithomimids [8, 19, 40, 41], but differs from other therizinosaurians, which possess relatively longer preacetabular portions of the ilium, including the earliest-branching member *Falcarius* and later-branching *Nothronychus* and *Segnosaurus* [19, 31].

**Equidimensional pubic peduncle of ilium.**   In *B. inexpectus*, the anteroposterior length and width of the public peduncle is approximately the same. This is a transitional feature linking early-branching to late-branching members, and although it is found in other coelurosaurs, it is currently unique among therizinosaurs. In earlier-branching therizinosaurians, such as in *Falcarius* and *Jianchangosaurus* ([19, 20], Fig 9), the anteroposterior length is greater than its width, whereas in later-branching members (e.g. *Suzhousaurus* [30], *Nanshiungosaurus* [22], and *Segnosaurus* [54]), the width is significantly greater than its length.

### New proposed synapomorphies for therizinosaurians

Here we discuss three possible synapomorphies found in this study—two for Therizinosauroidea and another for Therizinosauridae.

**Straight ulna.** In *B. inexpectus*, the ulna is straight as in most other therizinosaurian taxa (including *Jianchangosaurus*, *Alxasaurus*, *Nothronychus*, *Erliansaurus*, and *Therizinosaurus* [6, 20, 26, 32], but the ulna of the earliest-branching member *Falcarius* is bowed [2]. Therefore, a straight ulna appears to be a synapomorphy of Therizinosauroidea, and a reversal among coelurosaurs.

Among other coelurosaurian dinosaurs, a straight ulna is present in medium-bodied (e.g., *Sinornithomimus* [55]) and large-bodied taxa (e.g., some ornithomimids and *Gallimimus* [56], *Tyrannosaurus* [44], and *Gorgosaurus* [57]). In many smaller sized maniraptoran dinosaurs (e.g., *Microraptor* [47], *Heyuannia* [58], and *Confuciusornis* [59]), the ulna is bowed as in *Falcarius*.

**Blunt termini of furcula.** In *B. inexpectus*, *Falcarius* [19], and *Jianchangosaurus* [20], the furcular termini taper to a point. This is different from the robust and blunt termini in *Neimongosaurus* [23]. The blunt and robust condition may be a synapomorphy of Therizinosauridae. It is likely that the evolution of blunt termini has functional implications; however, further research is needed to understand the impact of this trait. Moreover, it is difficult to determine whether this feature is a synapomorphy for Therizinosauridae or just an autapomorphy for *Neimongosaurus* because currently the preservation of this element is rare among the clade.

**Broad intraclavicular angle.** As mentioned by Zanno [3], *Falcarius* possesses an acute angle (approximate 104 degrees) compared to *B. inexpectus* and *Neimongosaurus* (approximately 150 degrees). Therefore, a wider intraclavicular angle may be a synapomorphy for Therizinosauroidea.

## Conclusions

Since first reported in 1999 and 2003, the holotype of *B. inexpectus* has only been briefly described. In our restudy of this specimen, we not only provide a detailed description and illustrations of the known elements, but describe materials of the holotype that are not mentioned in previous papers (e.g., antero-middle dorsal vertebrae and middle dorsal vertebrae). Our morphological observations support previous proposals for the phylogenetic position of *B. inexpectus* as the earliest-branching member of Therizinosauroidea.

We also expand the diagnosis of this taxon by adding three additional autapomorphies of the postcranial skeleton (ungual II shorter than III, subequal length of the pre- and postacetabular, and equidimensional pubic peduncle of ilium). Furthermore, we describe new synapomorphies for more inclusive therizinosaur taxa.

Detailed information on the osteology of the postcranial skeleton of *B. inexpectus* is important for understanding the evolution of therizinosaurs since the majority of therizinosaur taxa are known exclusively from postcranial materials. The updated diagnosis and possible synapomorphies proposed in this study shed new light on the phylogeny and alpha taxonomy of Therizinosauria.

## Acknowledgments

The authors would like to thank Ding Xiaoqing for preparing the specimen and Qin Zichuan (University of Bristol) for valuable comments on the early versions of the manuscript. We are also grateful to two anonymous reviewers.

## Author Contributions

**Conceptualization:** Chun-Chi Liao, Xing Xu.

**Data curation:** Chun-Chi Liao, Shiying Wang.

**Formal analysis:** Chun-Chi Liao, Lindsay E. Zanno, Shiying Wang.

**Funding acquisition:** Xing Xu.

**Methodology:** Chun-Chi Liao, Lindsay E. Zanno, Shiying Wang, Xing Xu.

**Resources:** Lindsay E. Zanno, Xing Xu.

**Supervision:** Xing Xu.

**Validation:** Lindsay E. Zanno, Xing Xu.

**Visualization:** Chun-Chi Liao, Lindsay E. Zanno, Xing Xu.

**Writing – original draft:** Chun-Chi Liao.

**Writing – review & editing:** Chun-Chi Liao, Lindsay E. Zanno, Xing Xu.

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
