## [Decision Letter · Decision Letter 0]

14 Jul 2021

PONE-D-21-14221

Postcranial osteology of * Beipiaosaurus inexpectus *(Theropoda: Therizinosauria)

PLOS ONE

Dear Dr. Liao,

Thank you for submitting your manuscript to PLOS ONE. After careful consideration, we feel that it has merit but does not fully meet PLOS ONE’s publication criteria as it currently stands. Therefore, we invite you to submit a revised version of the manuscript that addresses the points raised during the review process.

Please take care of the suggestions of both reviewers (mainly improvements of writing).

We look forward to receiving your revised manuscript.

Kind regards,

Ulrich Joger

Academic Editor

PLOS ONE

Journal Requirements:

2. In your manuscript, please provide additional information regarding the specimens used in your study. Ensure that you have reported specimen numbers and complete repository information, including museum name and geographic location. 

For more information on PLOS ONE's requirements for paleontology and archaeology research, see https://journals.plos.org/plosone/s/submission-guidelines#loc-paleontology-and-archaeology-research

3. We note that you have stated that you will provide repository information for your data at acceptance. Should your manuscript be accepted for publication, we will hold it until you provide the relevant accession numbers or DOIs necessary to access your data. If you wish to make changes to your Data Availability statement, please describe these changes in your cover letter and we will update your Data Availability statement to reflect the information you provide

5. We note that Figure(s) 1, 2, 3, 4, 5, 6, 7, 8 and 9 in your submission contain copyrighted images. All PLOS content is published under the Creative Commons Attribution License (CC BY 4.0), which means that the manuscript, images, and Supporting Information files will be freely available online, and any third party is permitted to access, download, copy, distribute, and use these materials in any way, even commercially, with proper attribution. For more information, see our copyright guidelines: http://journals.plos.org/plosone/s/licenses-and-copyright.

1. You may seek permission from the original copyright holder of Figure(s) 1, 2, 3, 4, 5, 6, 7, 8 and 9  to publish the content specifically under the CC BY 4.0 license. 

Reviewers' comments:

Reviewer's Responses to Questions

**Comments to the Author**

1. Is the manuscript technically sound, and do the data support the conclusions?

Reviewer #1: Yes

Reviewer #2: Yes

2. Has the statistical analysis been performed appropriately and rigorously? 

Reviewer #1: Yes

Reviewer #2: N/A

3. Have the authors made all data underlying the findings in their manuscript fully available?

Reviewer #1: Yes

Reviewer #2: Yes

4. Is the manuscript presented in an intelligible fashion and written in standard English?

Reviewer #1: Yes

Reviewer #2: No

5. Review Comments to the Author

Reviewer #1: Dear authors,

indeed this is a very important manuscript due to the circumstance, that the description of the postcranial material of Beipiaosaurus inexpectus still was never published in the global community. The descriptive part of the manuscript is very informative and fits perfectly to the pictures. The coloration of the bones within the photographs sometimes helps, but sometimes leads to a very unnatural appearance of the image. Altogether a very interesting paper!

Reviewer #2: This is a solid paper, and a welcome addition to previously published information on Beipiaosaurus inexpectus. The osteological descriptions are generally good and the images are very helpful (recoloring the bones to highlight their morphology was a good idea).

The authors also provide a lot of contextual/comparative information and highlight areas of uncertainty (i.e. identity of carpals), so I do not have any major complaints or comments regarding the scientific merits of the paper.

The text needs a serious polish in terms of wording, grammar etc. but that should only take a couple of hours, so "minor revisions". I have highlighted some examples below:

I would reword some sentences in the abstract, for example:

Beipiaosaurus inexpectus, from the Lower Cretaceous Yixian Formation of Sihetun Locality near Beipiao

"Locality" should be lower case, and "from the Lower Cretaceous Yixian Formation (Sihetun locality, near Beipiao)" sounds better.

Page 3, lines 56, 61: "disentangle" would be a better word here.

Page 3, line 63: Is pygostyle supposed to be in quotation marks here?

Page 3, line 64: this should probably read "a unique type of feathers"

Page 3, line 65: Repetition of "of the"

In Table 1 (vertebral measurements), please use "vertebrae" (i.e., cervical vertebrae) because you are including measurements for several vertebrae.

Page 9, line 165 - this needs fixing: "there is a web of bone connected the two parts called intrapostzygapophyseal lamina

Page 11, line 212 - this needs fixing: "The well-developed lamina and fossa system is also a feature can be seen"

Page 11, line 223 - missing space: "heightas"

Page 12, line 237-239: missing parenthesis

Page 12, line 249: This should read "The last three sacrals"

Page 12, line 254: This should read subrectangular

Page 14, line 296 and throughout that paragraph: This should be the plural form (postzygapophyses)

Page 14, line 301: "most crushed"? Needs fixing.

Page 15, line 314: caudal series (not caudals series)

Table 2 - please check and fix the wording (e.g., "some midshaft might lost")

Page 17, line 335: remove "in"

Page 17, lines 338-340: reword this last sentence.

Page 17, line 341: repetition of "~"

Page 17, line 346: missing space

Page 18, line 362: this should read "similar to those"

Page 18, line 370: Should read "the glenoid"

Page 19, not sure about the use of "enhanced" here: "This feature may be enhanced by or entirely the result of crushing"

Page 21, line 436-438: Should read "the humeral head"

Page 21, line 442: Separate the words

Page 21, line 453: Should read "the humerus"

Page 22, line 476: This should read "There is a prominent..."

Page 22, line 483: This should be the plural form of radius...

Page 23, line 507: This first sentence is not really complete.

Page 25, please rewrite: "In general, metacarpal elements of B. inexpectus is resemble to those of Falcarius [2]."

Page 30, please reword: "The ilium is parallelogram in shape"

Page 33, line 740: Prominent

6. PLOS authors have the option to publish the peer review history of their article (what does this mean?). If published, this will include your full peer review and any attached files.

Reviewer #1: No

Reviewer #2: No

---

## [Author Response · Author response to Decision Letter 0]

16 Jul 2021

Dear Dr. Joger and anonymous reviewers,

We are thankful for your consideration of our manuscript and for the generous comments by the reviewers. We have updated the manuscript in light of the comments received, and believe that our revised submission is now suitable for publication.

Below, we enumerate the changes we have made to the manuscript in response to the requested revisions: 

1) In journal requirements, the permits are asked to state in manuscript. 

The specimen has been published before and housed in IVPP, so no field or related permits are needed in this study. We have added the statement in Methods (Lines 79-80).

2) Journal requirements indicates “We note that Figure(s) 1, 2, 3, 4, 5, 6, 7, 8 and 9 in your submission contain copyrighted images…”.

All the figures were drawn or token by authors of this article, so all of them are not under copyright. We have made a more detailed statement in “Copyrighted Figures” of “Additional Information”.

3) Reviewer 1 suggests explaining the reason of terminology choosing of manual digits (Lines 73-75).

The choosing of the terminology is to in light of the recent research of avian digit homology. We have added the reason in lines 75-76.

4) Reviewer 1 suggests using only genetic name or genetic and species name in lines 130-135.

We fixed it into genetic name only throughout the paragraph, and note the species name and number when referring to specific specimen.

5) Reviewer 1 suggests a more detailed description in line 328 instead of just saying “different”.

We have added a brief description of the chevrons of Falcarius and Alxasaurus to show how do they different from the chevrons of Beipiaosaurus (Lines 328-329).

6) Both reviewers found the sentence in lines 342-345 needs rewording.

We have miswritten the sentence and reworded it to make the contents more reasonable now (Lines 342-343).

And we have followed all the writing improvement suggestions made by both reviewers.

Many thanks for your consideration, and best wishes,

Chun-Chi Liao

On behalf of all authors.

---

## [Decision Letter · Decision Letter 1]

14 Sep 2021

Postcranial osteology of * Beipiaosaurus inexpectus *(Theropoda: Therizinosauria)

PONE-D-21-14221R1

Dear Dr. Liao,

We’re pleased to inform you that your manuscript has been judged scientifically suitable for publication and will be formally accepted for publication once it meets all outstanding technical requirements.

Kind regards,

Ulrich Joger

Academic Editor

PLOS ONE

Additional Editor Comments (optional):

Reviewers' comments:

Reviewer's Responses to Questions

**Comments to the Author**

1. If the authors have adequately addressed your comments raised in a previous round of review and you feel that this manuscript is now acceptable for publication, you may indicate that here to bypass the “Comments to the Author” section, enter your conflict of interest statement in the “Confidential to Editor” section, and submit your "Accept" recommendation.

Reviewer #2: (No Response)

2. Is the manuscript technically sound, and do the data support the conclusions?

Reviewer #2: Yes

3. Has the statistical analysis been performed appropriately and rigorously? 

Reviewer #2: N/A

4. Have the authors made all data underlying the findings in their manuscript fully available?

Reviewer #2: Yes

5. Is the manuscript presented in an intelligible fashion and written in standard English?

Reviewer #2: Yes

6. Review Comments to the Author

Reviewer #2: (No Response)

7. PLOS authors have the option to publish the peer review history of their article (what does this mean?). If published, this will include your full peer review and any attached files.

Reviewer #2: No

---

## [Editor Report · Acceptance letter]

20 Sep 2021

PONE-D-21-14221R1 

Postcranial osteology of *Beipiaosaurus inexpectus* (Theropoda: Therizinosauria) 

Dear Dr. Xu:

I'm pleased to inform you that your manuscript has been deemed suitable for publication in PLOS ONE. Congratulations! Your manuscript is now with our production department. 

Kind regards, 

on behalf of

Dr. Ulrich Joger 

Academic Editor

PLOS ONE